# Huanglongbing Pandemic: Current Challenges and Emerging Management Strategies

**DOI:** 10.3390/plants12010160

**Published:** 2022-12-29

**Authors:** Dilip Ghosh, Sunil Kokane, Brajesh Kumar Savita, Pranav Kumar, Ashwani Kumar Sharma, Ali Ozcan, Amol Kokane, Swadeshmukul Santra

**Affiliations:** 1Plant Virology Laboratory, ICAR-Central Citrus Research Institute, Nagpur 440033, India; 2Department of Biosciences and Bioengineering, Indian Institute of Technology Roorkee, Roorkee 247667, India; 3Vocational School of Technical Sciences, Karamanoglu Mehmetbey University, 70200 Karaman, Turkey; 4Scientific and Technological Studies Application and Research Center, Karamanoglu Mehmetbey University, 70200 Karaman, Turkey; 5Departments of Chemistry, Nano Science Technology Center, and Burnett School of Biomedical Sciences, University of Central Florida, Orlando, FL 32816, USA

**Keywords:** HLB pandemic, citrus greening, *Candidatus* Liberibacter asiaticus, a triangular disease management approach, integrated disease management

## Abstract

Huanglongbing (HLB, aka citrus greening), one of the most devastating diseases of citrus, has wreaked havoc on the global citrus industry in recent decades. The culprit behind such a gloomy scenario is the phloem-limited bacteria “*Candidatus* Liberibacter asiaticus” (*C*Las), which are transmitted via psyllid. To date, there are no effective long-termcommercialized control measures for HLB, making it increasingly difficult to prevent the disease spread. To combat HLB effectively, introduction of multipronged management strategies towards controlling *C*Las population within the phloem system is deemed necessary. This article presents a comprehensive review of up-to-date scientific information about HLB, including currently available management practices and unprecedented challenges associated with the disease control. Additionally, a triangular disease management approach has been introduced targeting pathogen, host, and vector. Pathogen-targeting approaches include (i) inhibition of important proteins of *C*Las, (ii) use of the most efficient antimicrobial or immunity-inducing compounds to suppress the growth of *C*Las, and (iii) use of tools to suppress or kill the *C*Las. Approaches for targeting the host include (i) improvement of the host immune system, (ii) effective use of transgenic variety to build the host’s resistance against *C*Las, and (iii) induction of systemic acquired resistance. Strategies for targeting the vector include (i) chemical and biological control and (ii) eradication of HLB-affected trees. Finally, a hypothetical model for integrated disease management has been discussed to mitigate the HLB pandemic.

## 1. Introduction

Citrus is the most widely grown specialty fruit crop in the world, containing a variety of health-promoting compounds, including vitamin C. The crop is highly vulnerable to various fungal, bacterial, and viral diseases, owing to its narrow genetic diversity [1]. Huanglongbing (HLB, aka citrus greening) is one of the most devastating diseases, which has affected the global citrus industry during last few decades [2,3]. The disease was first reported in southern China [4]. The discovery of HLB in India was attributed to a citrus dieback in the 1700s [5,6], resulting in a hypothesis that the disease was established in India before spreading to China [3,7]. A similar malady was observed in South Africa in 1929 and named “citrus greening disease” based on the poor color development of the stylar end of affected fruit [8]. The disease was also confirmed in South America, in the state of Sao Paulo in Brazil in 2004 [9], and in the state of Florida in the USA [10]. It has seriously impacted the US citrus industry, with an approximate loss of USD 3.6 billion per year [11]. In the USA, the disease was also detected in other states, including two significant citrus-producing states, Texas [12] and California [13], as well as in South Carolina, Georgia, and Louisiana [14]. The disease had also become established in several Caribbean countries such as Cuba [15], Jamaica [16], Belize [17], and Mexico [18]. Other major citrus-growing areas of the Mediterranean Basin and Australia are under threat. The disease also has moved west from Pakistan into Iran [19] and is threatening the neighboring areas. Presently, the disease is distributed in over 58 countries of Asia, America, Africa, Oceania, and the Caribbean. Reports are based on symptomatology, DNA-DNA hybridization with specific probe, PCR followed by *Xbal* restriction digestion of the amplified DNA, electron microscopy, and real-time PCR (Figure 1, Table 1) [20].

Typical symptoms of the disease are yellowing of shoots with mottled blotchy leaves (partly yellow/green, with several shades of yellow blending), corky veins, and green islands as depicted in Figure 2 [41,42]. The localized symptoms of greening eventually spread on the entire canopy and finally cause defoliation and tree dieback [43]. The symptoms of the chlorotic pattern often resemble zinc and iron deficiencies, as well as other diseases such as citrus tristeza, citrus stubborn, and phytophthora infection [44,45,46]. It is often seen that fruits from infected trees are small, lopsided, poorly colored, and bitter in taste (Figure 2). The root system is found to be underdeveloped due to starvation that leads to loss of fibrous roots [47]. All the species and hybrids of citrus, irrespective of their rootstock, are susceptible to the greening disease. However, symptoms vary from cultivar to cultivar, with the most severe found on sweet orange (*C. sinensis*), mandarin (*C. reticulata*), tangelo (*C. tangelo*), and grapefruit (*C. grandis*). Less severe symptoms are observed on lemon, rough lemon, and sour orange [2]. There are no known resistant citrus species for the disease, but some cultivars are more tolerant. For example, grapefruit is more tolerant than sweet orange. The pomelo (*Citrus maxima*) and kumquat (*Fortunella margarita*) cultivar were initially considered as tolerant but eventually became infected and started showing mottling symptoms [2,48].

## 2. Causative Agent, Genomics, and Pathogenesis Mechanism

The pathogen associated with HLB was initially thought to be a mycoplasma-like organism. Subsequent electron microscopic studies confirmed that the causative organism is a bacterium. The fastidious nature of the pathogen was an impediment in traditional taxonomical classification like the study of morphology and growth characteristics. The phloem-limited causal agent was classified based on the 16S rRNA gene sequence and grouped under the α-subdivision of proteobacteria, genus *Candidatus* Liberibacter in the family *Rhizobiaceae* [49,50]. So far, three species of bacterium are known to be associated with citrus greening disease: ‘*Candidatus* Liberibacter asiaticus’ (*C*Las), ‘*Candidatus* Liberibacter africanus’ (*C*Laf), and ‘*Candidatus* Liberibacter americanus’ (*C*Lam). To date, no successful attempts have been made to grow these bacteria in culture.

Among them, *C*Las is the most destructive, widely prevalent, highly divergent, and has caused significant economic loss in citrus production globally [51]. *C*Lam and *C*Laf are only present in Brazil and Africa, respectively. *C*Lam, originally identified in Brazil, was the major species, but later *C*Las became the most prevalent species [52]. This intracellular plant pathogen acts as an insect symbiont and is transmitted by two sap-sucking insect species, *Diaphorinacitri* and *Triozaerytreae*. *D. citri* isalso known as the Asian citrus psyllid (ACP) (Figure 3). The ACP is responsible for the spread of *C*Las and *C*Lam in Asia as well as in the Americas [53]. The ACP is heat-tolerant and can withstand high temperatures (up to 45 °C) but is sensitive to high humidity (above 90%) [54]. On the other hand, *T. erytreae,* African citrus psyllid (AfCP), the vector for spread of *C*Laf in Africa [29], is heat-sensitive. The adult and juvenile forms grow in a cool, moist environment and cannot withstand temperatures above 32 °C [55]. The rapid spread of HLB throughout the globe sparked research interest in understanding the genomics, transcriptomics, and proteomics of the host/vector/pathogen virulence and diversity.

Despite the unculturable nature of *C*Las, the complete circular genome sequence was generated from *C*Las-infected psyllid by metagenomics, which became the foundation platform for further research in functional genomics [56]. Presently, 42 complete isolate sequences of *C*Las are available in GenBank.Ten genomes are fully assembled in a single scaffold: psy62 [56], gxpsy [57], Ishi-1 [58], A4 [59], JXGC [60], AHCA1 [61], JRPAMB1 [62], TaiYZ2 [63], CoFLP1 [64], and ReuSP1 [65] (Table 2). A total of 4.5% and 8% of genes are involved in cell motility and active transport mechanism, respectively, and they might contribute to its virulence activity in the citrus plant phloem system [56]. Bacterial plant pathogens use the secreted proteins (effectors) in their defense mechanism to suppress plant immunity and create a favorable environment for colonization and proliferation [66,67]. The *C*Las genome consists of all Type I secretion system genes that encode proteins involved in multidrug efflux and toxin effectors: HlyD (membrane fusion protein, CLIBASIA_01355), PrtD (ABC transporter, CLIBASIA_1350), and TolC (outer membrane export protein, CLIBASIA_04145). However, *C*Las lacks type III, type IV, and type VI secretion systems and typical degradative enzymes, which are required for its free-living state [67,68,69].

*C*Las also possesses the general Sec secretion system/Sectransloconcapable of pathogenicity to the host plants by secreting effectors directly outside bacterial cells. *C*Las secretory proteins CLIBASIA 05315, CLIBASIA 03875, CLIBASIA 00460, and CLIBASIA 04025 have been reported as Sec-dependent secretory proteins engaged in starch accumulation, cell death, and host plant infections [67,68,69,70,71]. Different peroxidase enzymes, such as SC2_gp095 and CLIBASIA_RS00445, have been identified as non-classical secretory proteins in *C*Las, which counter the reactive-oxygen-species (ROS)-mediated defense-signaling response, including H_2_O_2_, used by plants to combat disease progression [69]. This indicates that *C*Las may have developed a non-classical secretion pathway to release virulence proteins to combat the host. According to secretome analysis, the *C*Las genome contains a total of 27 non-classically secreted proteins (ncSecPs), the majority of which are involved in suppressing early plant defense mechanisms by diminishing the hypersensitive response [69]. The peroxiredoxin (Prx) superfamily proteins are ubiquitous cysteine-based non-heme peroxidases present in *C*Las. For example, bacterioferritin comigratory protein (BCP) is involved in the oxidative stress defense system of *C*Las due to its ROS scavenging activity [72]. Lipopolysaccharides (LPS), the most important outer membrane module of *C*Las encoded by 21 genes, not only play a critical role in maintaining the robust structural integrity to the bacterial cell, but also play a role in the virulence mechanism. However, there are some differences between *C*Las, *C*Laf, and *C*Lam for type I secretion system, and LPS production has been reported [67,73].

Quorum sensing is a cell-to-cell signaling cascade where chemical-based regulatory communications occur among bacterial populations for their motility, biofilm formation, and virulence mechanism [11]. The mechanism of quorum sensing is regulated by two genes: *luxI* and *luxR*. The *luxI* gene encodes different quorum-sensing molecules, acyl-homoserine lactone (AHL), which induce biofilm formation by activation of *luxR* genes [68]. As *C*Las has a solo LuxR system but lacks LuxI [56], there is currently no evidence on how the *C*Las pathogen employs a quorum-sensing-based mechanism to cause the pathogenicity in citrus plants, although it is speculated that the disease is established like other phytopathogens [74,75]. It has been hypothesized that the communication among the *C*Las, endosymbiont, and psyllid is based on luxR and luxl genes [74]. *C*Las potentially communicates with the endosymbiont (*Wolbachia* spp.) and psyllid after adhering in the saliva sheath. Proteins like Mucin-5AC protein (23.46 kDa) were identified in *D. citri* saliva in a proteome study, which might be involved in the formation of the salivary sheath. Studies have shown that the down-regulation of Mucin 5AC results in reduced bacterial pathogen acquisition by inhibiting bacterial adhesion to the insect gut [76]. It has been reported that some proteins of psyllid (haemocyanin protein and myosin protein) and *C*Las (phosphopantothenoylcysteine synthetase and pantothenate kinase) interact with each other after the acquisition of *C*Las [76]. Therefore, a comprehensive understanding of the quorum-sensing system in *C*Las and the interaction with the citrus and the vector with respect to co-evolved protein interaction networksmay provide a target for combating HLB by hampering acquisition, growth, and biofilm formation of *C*Las.

**Table 2 plants-12-00160-t002:** Details of sequenced genomes of *C*Las, *C*Lam, and *C*Laf.

Sr. No	*Candidatus* Liberibacter spp.	Strain	Host	Sample Origin	Genome Size (Mb)	Number CDS Present in the Genome	Reference
1	*Candidatus* Liberibacter asiaticus	A4 (CP010804.1)	*Citrus reticulata*	China: Guangdong	1.23025	1067	[59]
2	*Candidatus* Liberibacter asiaticus	Gxpsy (CP004005.1)	*Diaphorinacitri*	China: Guangxi	1.26824	1094	[57]
3	*Candidatus* Liberibacter asiaticus	JRPAMB1 (CP040636.1)	*Diaphorinacitri*	USA: Florida	1.23716	1072	[62]
4	*Candidatus* Liberibacter asiaticus	TaiYZ2 (CP041385.1)	*Citrus maxima*	Thailand: Songkhla	1.23062	1067	[77]
5	*Candidatus* Liberibacter asiaticus	psy62 (CP001677.5)	-	USA: Florida	1.22732	1049	[56]
6	*Candidatus* Liberibacter asiaticus	JXGC (CP019958.1)	Citrus	China: Jiangxi	1.22516	1033	[60]
7	*Candidatus* Liberibacter asiaticus	Ishi-1 (AP014595.1)	*Diaphorinacitri*	Japan: Ishigaki	1.19085	1001	[58]
8	*Candidatus* Liberibacter asiaticus	AHCA1 (CP029348.1)	*Diaphorinacitri*	USA: California	1.23375	1056	[61]
9	*Candidatus* Liberibacter asiaticus	FL17 (JWHA00000000.1)	Citrus	USA: Florida	1.22725	1019	[78]
10	*Candidatus* Liberibacter asiaticus	YNJS7C (QXDO00000000)	Citrus	China: Yunnan	1.25898	1102	[79]
11	*Candidatus* Liberibacter asiaticus	YCPsy LIIM00000000)	*Diaphorinacitri*	China: Guangdong	1.233647	1037	[80]
12	*Candidatus* Liberibacter asiaticus	LBR19TX2 (VTMA00000000	-	USA: Texas	1.20275	1008	[67]
13	*Candidatus* Liberibacter asiaticus	LBR23TX5 (VTMB00000000)	-	USA: Texas	1.20347	1009	[67]
14	*Candidatus* Liberibacter asiaticus	AHCA17 (VNFL00000000)	Citrus maxima	USA: California	1.20862	1036	[81]
15	*Candidatus* Liberibacter asiaticus	YNXP-1 (VIGA00000000)	Cuscuta	China: Yunnan	1.20707	1031	-
16	*Candidatus* Liberibacter asiaticus	SGCA16 (VTLZ00000000)	-	USA: San Gabriel	1.20994	1015	[67]
17	*Candidatus* Liberibacter asiaticus	JXGZ-1 (VIQL00000000)	Cuscuta	China: JiangXi	1.21799	1040	-
18	*Candidatus* Liberibacter asiaticus	DUR1TX1 VTLT00000000.1)	-	USA: Texas	1.20629	1011	[67]
19	*Candidatus* Liberibacter asiaticus	Mex8 (VTLU00000000.1)	-	Mexico: Mexicali	1.24313	1042	[67]
20	*Candidatus* Liberibacter asiaticus	SGCA5 (LMTO00000000.1)	Orange citrus	USA: San Gabriel	1.20138	1001	[80]
21	*Candidatus* Liberibacter asiaticus	CHUC (VTLV00000000)	-	China	1.20845	1032	[67]
22	*Candidatus* Liberibacter asiaticus	TX2351 (MTIM00000000)	Asian citrus psyllid	USA: Texas	1.252	1129	[82]
23	*Candidatus* Liberibacter asiaticus	GFR3TX3 (VTLR00000000)	-	USA: Texas	1.20932	1013	[67]
24	*Candidatus* Liberibacter asiaticus	HHCA16 (VTLY00000000)	-	USA: Hacienda Heights	1.20705	1012	[67]
25	*Candidatus* Liberibacter asiaticus	MFL16 (VTLX00000000)	-	USA: Florida	1.19922	1012	[67]
26	*Candidatus* Liberibacter asiaticus	DUR2TX1 (VTLS00000000)	-	USA: Texas	1.21232	1009	[67]
27	*Candidatus* Liberibacter asiaticus	CRCFL16 (VTLW00000000)	-	USA: Florida	1.20828	1028	[67]
28	*Candidatus* Liberibacter asiaticus	HHCA (JMIL00000000.2)	*Citrus* sp.	USA: Hacienda Heights	1.15062	867	[59]
29	*Candidatus* Liberibacter asiaticus	TX1712 (QEWL00000000)	*Citrus sinensis*	USA: Texas	1.20333	0	[83]
30	*Candidatus* Liberibacter asiaticus	SGpsy (QFZJ00000000.1)	*Diaphorinacitri*	USA: San Gabriel	0.769888	0	[61]
31	*Candidatus* Liberibacter asiaticus	SGCA1 (QFZT00000000.1)		USA: San Gabriel	0.233414	557	[61]
32	*Candidatus* Liberibacter asiaticus	YCPsy (LIIM00000000)	*Diaphorinacitri*	Guangdong, China	1.233647	-	[80]
33	*Candidatus* Liberibacter asiaticus	PA19 (WOXD01000000)	Kinnow mandarin	Pakistan	1.224156	1059	[84]
34	*Candidatus* Liberibacter asiaticus	PA20 (WOUN01000000)	Kinnow mandarin	Pakistan	1.226225	1062	[84]
35	*Candidatus* Liberibacter asiaticus	CoFLP1 (CP054558.1)	*Diaphorinacitri*	Colombia: Municipio Dibulla	1.231639	1048	[64]
36	*Candidatus* Liberibacter asiaticus	9PA (JABDRZ000000000.1)	*Citrus sinensis*	Brazil (South America)	1.231881	-	[85]
37	*Candidatus* Liberibacter asiaticus	MFL16 (VTLX00000000)	Citrus	USA: Florida	1,199,225 bp	-	[67]
38	*Candidatus* Liberibacter asiaticus	CRCFL16 (VTLW00000000)	Citrus	USA: Florida	1,208,280 bp	-	[67]
39	*Candidatus* Liberibacter asiaticus	ReuSP1 (CP061535.1)	*Diaphorinacitri*	France: La Reunion	1.230064	1043	[65]
40	*Candidatus* Liberibacter asiaticus	Tabriz.3(JAKQYA000000000.1)	*Elaeagnus angustifolia*	Iran: East Azerbaijan, Tabriz	1.22409	589	-
41	*Candidatus* Liberibacter asiaticus	YNHK-2 (WUUB01000000.1)	Citrus	China: Yunnan	1.08957	-	[86]
42	*Candidatus* Liberibacter asiaticus	A-SBCA19 (JADBIB010000000.1)	*Diaphorinacitri*	USA: California, San Bernardino County	1.18688	1067	[87]
43	*Candidatus* Liberibacter americanus	Sao Paulo (NC_022793)	*Citrus sinensis*	Brazil	1.1952	945	[73]
44	*Candidatus* Liberibacter americanus	PW_SP	Catharanthus roseus	Brazil: Sao Paulo	1.17607	924	-
45	*Candidatus* Liberibacter africanus	PTSAPSY	Psyllid	South Africa: Pretoria	1.19223	1036	-

### Pathogen Virulence Factors

Recent studies have put emphasis on understanding the virulence mechanisms of *C*Las in the citrus host. The contribution of prophages in *C*Las pathogenicity towards the suppression of plant defense has been reported [88]. Initially, it was reported that *C*Las bacterium carries two prophages, Type 1 (SC1) and Type 2 (SC2). Recently, the prophages have been classified into three types, i.e., Type 1 (SC1), Type 2 (SC2), and Type 3 (P-JXGC-3), based on functional and comparative genomic analysis of 15 different *C*Las genomes [89]. The SC1 prophage is reported tobe lytic as it produces proteins necessary for the lytic cycle and becomes replicative in plants [90]. Phage particles were observed in the phloem of infected periwinkle and sweet orange plants [91]. SC2, on the other hand, is a replicative excision plasmid that lacks lytic genes and may play a role in the lysogenic cycle. SC2 encodes proteins, i.e., peroxidase (SC2_gp095) and glutathione peroxidase (SC2_gp100); it has been observed that transient expression of SC2 gp095 leads to suppression of H_2_O_2_-mediated defense signaling in plants [88,90]. The *C*Las may use the peroxidase enzyme as a defense mechanism against the host immune response by suppressing the plant’s H_2_O_2_-mediated hypersensitive response [88,90]. Zheng et al. (2016) studied the dominating *C*Las strains in southern China, revealing a single prophage, SC1 (90.4%) or SC2 (82.6%), over other strains [92]. The in silico analyses of CGdP2 have identified the presence of CRISPR/cas systems in SC1 and SC2 prophages. Based on this analysis, it was hypothesized that the presence of a CRISPR/cas system in dominating species allows them to overcome an invading phage/prophage into the *C*Las genome. *C*Las also contains other virulence factors, like serralysin (CLIBASIA_01345) and hemolysin (CLIBASIA_01555). Serralysin is a metalloprotease, which inactivates various antimicrobial proteins involved in the plant defense mechanism. This enzyme is believed to be used by the *C*Las to defend against the citrus immune response [93]. To promote virulence, the endosymbiont-like pathogen ‘*Ca*. L. psyllaurous’ suppresses the expression of genes involved in the plant defense mechanism, i.e., genes regulated by jasmonic acid (JA) and salicylic acid (SA), by introducing protein effectors [94]. *C*Las also degrades SA, which plays a critical role in the plant defense mechanism against pathogens using salicylate hydroxylase.Salicylate hydroxylase reduces the defense action of the citrus plant by attenuating the response to exogenous SA [95]. The secretion and transport of the effector proteins in the host plant cells is one of the most important virulence factors of the bacterial pathogen [66]. Thevirulence factor CaLas5315 (Sec-delivered effector 1) hinders the papain-like cysteine protease’s activity to suppress the defense mechanism of citrus. It also induces the callose deposition inside the vascular tissue, starch formation, chlorosis, and plant cell death after localization in the chloroplast of *Nicotiana benthamiana* [69,71,96]. Ying et al. (2019) have assessed 60 total putative virulence factors of *C*Las and identified four candidates (detrimental virulence factors) which are responsible for growth inhibition (CLIBASIA_00470 and CLIBASIA_04025) and cell death (CLIBASIA_05150 and CLIBASIA_04065C) in *N. benthamiana* [97].

## 3. HLB Diagnosis

To combat the HLB pandemic, early disease diagnosis is important to minimize further damage to the global citrus industry. It is often challenging to visually distinguish HLB in the field from similar non-HLB-related symptoms which may be indicative of other ailments, such as citrus tristeza or nutrient deficiency [46,98]. Over the years, several techniques have been developed for HLB diagnosis, as discussed below.

### 3.1. Electron Microscopy

Electron microscopy was the first laboratory technique used to identify the pathogen [99]. Cevallos-Cevalloset al.used the transmission electron microscopy (TEM) technique to investigate a thin section of samples collected from leaf, petiole, stem, bark, and root tissue of an HLB-suspected plant and directly confirmed the presence of the pathogen [100]. TEM sample preparation involved the following steps: (i) tissue samples were fixed using 3% glutaraldehyde for 4 h at room temperature followed by overnight storage in 0.1 mol/L potassium phosphate buffer (pH 7.2) in a refrigerator, (ii) samples were subsequently washed in the same buffer and treated with 2% osmium tetroxide solution for 4 h at room temperature, (iii) samples were then dehydrated with acetone, cut into 90–100 nm sections, stained with 2% uranyl acetate (aqueous), and (iv) lastly, samples were post-stained in lead citrate and examined using a Morgani 268 TEM (Figure 4A).

### 3.2. Molecular-Based Assays

Molecular techniques are sensitive and reliable tools for plant disease diagnosis. The most commonly used techniques are polymerase chain reaction (PCR), real-time PCR (q-PCR), flow cytometry, fluorescence in situ hybridization (FISH), and DNA microarrays [101,102,103]. In conventional PCR, 16S-rRNA-sequence-specific primer sets OI1/OI2c and OA1/OI2c were used for *C*Las and *C*Laf detection, respectively [104] (Figure 4B). Nucleotide sequence analysis of the 16S rRNA region of both the species reveals that *C*Las has only one *Xba1* restriction site (this produced two DNA fragments of size 520 bp and 640 bp after restriction digestion). On the other hand, *C*Laf has two *Xba1* restriction sites, yielding three DNA fragments of a size of 520 bp, 506 bp, and 130 bp [49]. Another important genomic locus used for HLB diagnosis is the rplKAJL-rpoBC operon. The primer set f-rplA2/ r-rplJ5 specific to this region amplifies a 703 bp amplicon for *C*Las and 669 bp amplicon for *C*Laf, respectively. The *C*Lam is detected by another set of primers, i.e., f-GB1/r-GB3, which is specific to the 16S rRNA region of the *C*Lam [42].

qPCR is one of the most sensitive and reliable quantitative methods for gene expression analysis as well as for pathogen detection. Therefore, qPCR has become the preferred method for *C*Las detection [105] (Figure 4D), as it is capable of reliably detecting the pathogen at a very low concentration. Different chemistry/reporter dyes have been used in qPCR to improve the pathogen detection limit. Nageswara-Rao et al. (2013) had developed qPCR using various candidate genes for the early detection of HLB disease [42]. Ghosh et al. (2018) standardized qPCR with TaqMan chemistry to detect and evaluate the efficacy of an antimicrobial nano-zinc oxide-2S albumin protein formulation on the growth of *C*Las *in planta* [106].

Dot hybridization assay with a biotin-labeled DNA probe was successfully used for HLB diagnosis in various citrus hosts, including mandarins, tangors, sweet oranges, andpomelos. A nucleic acid spot hybridization (NASH) test was also developed for the diagnosis of HLB which could detect up to 1:100 dilution in HLB-infected tissue [107]. DNA microarray is an advanced molecular technique mostly used for transcriptome analysis in various bacterial plant diseases and has been used in the transcriptional profiling of sweet orange plants in response to infection with *C*Las using the Affymetrix GeneChip citrus genome microarray [108].

Loop-mediated isothermal amplification(LAMP) is another molecular technique where DNA amplification is carried out at isothermal temperature [109]. The *Bst* polymerase enzyme with a strand displacement property is used to complete the isothermal amplification reaction. This biochemical reaction is carried out using a simple water bath operating between 60 °C and 65 °C [110].

The LAMP method combined with a lateral flow dipstick (LFD) was also developed for rapid HLB diagnosis [111]. Ghosh et al. (2016) developed a rapid and sensitive LAMP technique using SYBR green I dye for visual *C*Las detection (Figure 4C) [112].

The recombinase polymerase amplification (RPA) is an isothermal nucleic acid amplification technique which requires three enzymes (a recombinase, a strand-displacing DNA polymerase, and a single-strand binding protein) for the extension of primers, induced by the recombination process. The reaction could be performed at isothermal temperature ranges between 37 °C and 42 °C within 15–25 min. Due to simple reaction conditions, the RPA is considered as one of the most promising emerging molecular diagnostic technologies. The *C*Las was also diagnosed with an RPA assay based on SYBR green I dye using a mini-UV torch light [113] (Figure 4E) andlateral flow assay (HLB-RPA-LFA) [114] (Figure 4F).

**Figure 4 plants-12-00160-f004:**
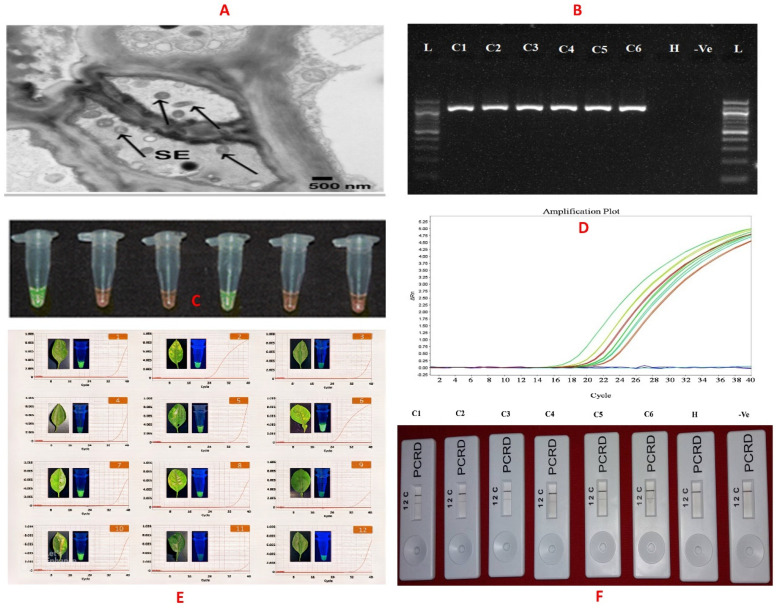
Different methods of *C*Las detection. (**A**) Transmission electron micrographs of *C*Las in a sieve element of a young leaf [115]. (**B**) PCR products visualized on agarose gel using primer set OI1/OI2C [L: 1kb DNA ladder, C1 to C6: greening-infected samples, H: healthy control, −Ve: negative control [114]. (**C**) LAMP-based detection using CYBR green I dye under normal light [tubes 1 and 3: greening-infected samples, tubes 2, 3, and 5: healthy control, and tube 6: non-template control [112] (**D**) TaqMan-qPCR with HLBas-F/R-HLBp primer probe pair. Amplification plot for sample C1 to C6 represent the *C*La*s*-positive sample. (**E**) HLB-positive leaf samples and their detection results by real-time PCR and RPA-based visual detection method [113]. (**F**) HLB-RPA-LFA [C1 to C6 represents the *C*Las isolates, H: Healthy control and−Ve: negative control [114].

### 3.3. Early Diagnosis

It has been documented that the *C*Las pathogen enters the healthy citrus tree via different mediators and spreads erratically in the vascular system [3]. Despite the high sensitivity and reliability, *C*Las detection methods such asPCRandqPCRsuffer from serious limitations for early diagnosis on a large scale. Molecular techniquesaretime-consuming and require laboratory setup and skilled technicians to run the assay. Therefore, these assays are not cost-efficient and are difficult to implement as a point-of-care tool for citrus growers. *C*Las is reported to be unevenly distributed in the citrus vascular system [3]. The likelihood of obtaining falsenegative results is therefore high even though the pathogen translocates and multiplies over time [3]. Trees at the early stage of infection with low *C*Las titer serve as an inoculum source for vector-assisted transmission to surrounding healthy trees. To meet the above needs, researchers are currently focusingon the development ofadvanced approaches for rapid HLB diagnosis that can be implemented in commercial orchards at a large scale, i.e., tens to hundreds of acres. One of the most promising research concepts is the profiling/screening of disease-specific volatile organic compounds (VOCs) released by the affected plant. The idea is to identify uniqueVOCprofiles/fingerprints in diseased plants that can be coupledwith an electronic odor detection system (also known as an electronic nose or E-nose). It is expected that a healthy plant will exhibit different VOCprofiles/fingerprints, thus differentiating it from the disease-affected trees. The E-nose system consists of a series of gas sensors with specific sensitivities to one or moreVOCs, generating a profiles/fingerprints to discriminate a mixture of different compounds present in the abnormal or healthy samples [116]. The mammalian olfactory system consisting of chemotacticreceptorshas been utilizedas a first aid to gauge the freshness, quality, and edibility of food products [117,118]. Canines possess a powerful olfactory system. Recently, canine olfactory surveillance was deployed to recognize the volatile chemicals generated byHLB-affected trees. It has been reported that canine accuracy detected 99% of HLB-affected trees (Figure 5). A likelihood of a 100% infection rate was achieved when two or more dogs were used for the same tree. The accuracy, sensitivity, and specificity of 10 dogs trained toidentify *C*Laswere 0.9905, 0.8579, and 0.9961, respectively [118]. Upon training, canines only reacted to *C*Lasand not to other citrus pathogens such as bacterial, viral, orspiroplasma [118].

## 4. Triangular Management Strategies

To mitigate the effects of HLB, there is a need to devise integrated management strategies which could obstruct the disease spread in a citrus grove. To date, no effective chemical control measures are available for HLB management, and therefore, it is becoming an increasingly difficult task to stop the spread of disease in new citrus-producing areas. Integrated disease management viaintervention at three different levels, pathogen, vector, and host, either individually or in combination, would be the most promising approach to combat HLB. Currently available management options include control of psyllid population chemically or biologically, removal of *C*Las-infected trees, and planting of disease-free nursery trees. To identify innovative HLB management options, it is important to understand pathogen biology, the pathogenesis mechanism, disease infection progression, and its correlation with genomics and proteomics. This article is focused on a comprehensive review of scientific information available to date concerning HLB control based on the triangular disease management approach involving pathogen, vector, and host (Figure 6).

### 4.1. Disease Control Strategies at Pathogen Level

The disease management strategies are mostly dependent on the control of the causal agent of the disease. Unfortunately, the causal agent of HLB is one of few phloem-limited pathogens that have not been grown in an axenic culture. Numerous efforts have been made globally to isolate and culture *C*Las. To date, there is no protocol available to culture *C*Las that is robust enough to be useful in supporting efficacy screening of bactericide candidates. The media composed of citrus vein extract (Liber A medium) and citrus juice (King’s B medium) could grow *C*Las colonies on agar plates. However, this media failed to stabilize the *C*Las for more than 4–5 serial single-colony transfers for further research use [11]. Successful culture of *C*Las strain Ishi-1 on a solid medium through a mutualistic relationship with citrus phloem microbiota has been also reported recently. However, this media failed to support culture of the other strains of *C*Las [119]. According to genome analysis, *C*Las lack genes that code for essential enzymes/other proteins and thus requires association of other citrus phloem microbiota for their survival. Inability to culture *C*Las seriously limits the development of effective long-term control strategies. To combat HLB, there is a desperate need for thedevelopment of strategies aimed towards inhibition of *C*Las multiplication inside the phloem system. Thus, to alleviate the effects of this graft-transmissible and systemic pathogen and protect the farmers’ interest, there is an urgent need to identify or develop novel inhibitor antimicrobial molecules that can suppress or eradicate these pathogens from the infected trees. Availability of genome information of *C*Las makes it easier to select key proteins critical for survival of the bacteria andthe potential inhibitory molecules against *C*Las.

#### 4.1.1. Protein-Based Approach/Targeting Important Proteins of *C*Las

A combination of the antibiotics penicillin and streptomycin has been illustrated to be effective in suppressing *C*Las [120]. Furthermore, several antibiotics like ampicillin, cefalexin, penicillin, carbenicillin, rifampicin, and sulfadimethoxine were found to be effective in suppressing *C*Las infection [121]. Proteins are the most versatile biological macromolecules in living systems and perform a diverse array of functions in essentially all biological processes. They function as transporters and catalysts, provide mechanical support and immune protection, transmit nerve impulses, store other molecules such as oxygen, and control growth and differentiation. Therefore, other potential strategies for controlling HLB disease include targeting essential proteins that are critical for the survival of bacteria through the development of inhibitor molecules to impair the target protein function [122]. The essential protein of *C*Las can be targeted with various inhibitor molecules based on two criteria, i.e., selectivity and necessity for virulence.

Transport protein

All living organisms import diverse nutrients from the environment and expel toxic elements and waste products outside the cell with the help of different membrane-embedded transporters [123]. The ABC (ATP-Binding Cassette) transporters are one of the important and common protein families among living organisms, from microorganisms to humans, that are involved in the movement of solutes across the cell membrane [124]. Different types of ABC transporters are found in bacteria, which are complexes of the transmembrane and solute-binding domain (SBD) that facilitate the unidirectional movement of extracytosolic molecules into the cytoplasm against their concentration gradient by the hydrolysis of the ATP molecule. The ABC transporters consist of two transmembrane domains (TMDs) that form a specific ligand transport channel and two cytosolic ATP-binding domains (ABDs) that hydrolyze ATP to provide the energy for the translocation of substrate across the membrane. The prokaryotic ABC transporter has periplasmic solute-binding protein (SBP), which traps the substrate in the periplasmic space and delivers it to the external surface of the transport complex (Figure 7).

ABC transporters import small molecules into the cell in association with the SBP, including sugars, amino acids, peptides, phosphate esters, inorganic phosphate, sulfate, phosphonates, metal cations, iron-chelator complexes, vitamins, and polyamines [125]. SBPs are involved in substrate identification, capture, and release to the translocator of the ABC importer. L-arabinose-binding protein, an SBP, was first identified and crystallized from the periplasmic space of *E. coli* [126]. SBPs have extremely conserved structural folds consisting of two globular domains (bigger and smaller domain/lobe), connected by a hinge region. Both bigger and smaller domains are built by a usual α/β fold with internal core β-sheets surrounded by α-helices. In the absence of substrate (open state), the two domains are well separated and rotating around the hinge region, and after substrate binding, they go to the closed state with major conformational changes [127,128]. It rotates around the hinge region; eventually, the two domains move towards each other and engulf the substrate like a “Venus Flytrap” [129]. Based on structural similarity and pairwise structural alignment, SBPs were classified into seven different clusters (A–G). On the basis of substrate specificity, four of these, “cluster A, B, D, and F”, were further subdivided [130]. These SBPs have evolved to recognize a wide variety of solutes with high affinity and specificity, and have also been involved in signal transduction, transcriptional regulation, and catalysis.

The Cluster F–IV and A–I families of substrate-binding proteins are involved in the transport of amino acids (cystine, cysteine, arginine, glutamine, histidine, glutamate/aspartate) and divalent metal ions (Zn^2+^, Mn^2+^, and Fe^2+^), respectively. The binding studies of Cluster F–IV and A–I show different substrate specificity (Table 3). The amino acid transporters are associated with a bacterial infection. The deletion of amino acid receptors reduces the virulence of bacteria. A recent study in *Moraxellacatarrhalis* reveals that the deletion of lysine and ornithine receptors reduces the invasion of host adenocarcinoma epithelial cells [131]. In the human pathogenic bacteria *Listeria monocytogenes*, deletion of the cysteine receptor (CtaP) results in increased acid sensitivity, membrane permeability, reduced bacterial adherence, and lowered colonization [132]. In *Salmonella typhimurium,* expression of the D-alanine ABC receptors (DalS) protects the bacteria from oxidative damage elicited by D-amino acid oxidase (DAO) in the host’s neutrophils. The deletion of DalS makes the bacteria more prone to DAO-dependent killing inside the host cell [133]. The inhibitors specific for solute-binding protein/receptors have proven to be a drug candidate. The compounds, RDS50 and RDS51, bind with the ZnuABC receptor of *S. typhimurium* and were shown to inhibit the growth of the pathogen. The crystal structure of the “RDS51-Zn(II)-ZnuA complex” has revealed that it binds near the zinc-binding site of the SBPs [134]. The *C*Las genome encodes 137 transporter proteins, which is an exceptionally high number compared to other intracellular bacteria with similar genome size. Among one hundred thirty-seven transporter proteins, twenty-four proteins are electrochemical-potential-driven transporters, nine proteins are channels/pores transporters, ninety-two proteins are primary active transporters, one protein is a translocator, and the remaining eleven proteins are uncharacterized transporters. Among 92 primary active transporters of *C*Las, 40 are ABC transporters [56,121]. The expression studies of some vital component of the ABC transporter complex changed due to switching of the host from psyllid to plant (Table 4) [135]. Some transporter proteins from *C*Las have been biochemically and biophysically characterized: ZnuABC, the zinc uptake system, the amino acid transporter, ATP/ADP translocase, and NttA. It is reported that the zinc transport system (*znuABC*) is associated with pathogenesis in bacteria [136]. It is possible that some of these transporters are involved in the uptake of nutrients and the virulence mechanism of *C*Las.

##### Znu System

Zinc is the cofactor and structural element of various proteins present in bacteria which is imported through the zinc uptake (Znu) system. The periplasmic domain ZnuA of the ZnuABC complex binds zinc in the periplasm and comes in close contact with theZnuBC complex and transports it into the cell. The *C*Las genome encodes two ZnuABC homologous systems; among them only one (ZnuA1) is functional and able to complement the function of the partially inactivated ∆ZnuA of *S. meliloti* and *E. coli*. It is assumed that the second (ZnuA2) of the two homologous systems might be involved in Mn^2+^ uptake, and therefore it does not complement the ∆ZnuA of *S. meliloti* and *E. coli* [122]. *S. meliloti* and *E. coli*, havea regulator of the Znu system, i.e., the Zur protein, but in *C*Las the gene encoding the homologue to this protein is absent, indicating the alternative mode of regulation in *C*Las [122]. The ZnuA1 of *C*Las having conserved metal-coordinating residues (3His and 1Asp), which are specific for Zn^2+^ binding, clearly indicated its role in Zn-uptake. ZnuA1 of *C*Las shared only 22% sequence identity with Clas-ZnuA2 and contained four conserved metal-coordinating residues (His39, His106, Glu 172, and Asp 247), which are known to be specific for Mn^2+^ or Fe^2+^ binding [147]. The sequence analysis of ZnuA2 showed the highest identity with the Cluster A-I family’s Mn^2+^/Fe^2+^-specific SBPs. The comparison of CLas-ZnuA2’s structure in three states (metal-free, intermediate, and metal-bound state) showed that the mechanism of metal binding resembles the Zn^2+^-specific SBPs of the A-I family [147]. Crystal structure showed that CLas-ZnuA2 binds both Zn^2+^ and Mn^2+^ with square pyramidal geometry, different from Mn-specific SBPs (tetrahedral geometry). The crystal structure studies of CLas-ZnuA2 (S38A and Y68F) protein demonstrate thatit is mutant in a metal-bound and metal-free states, confirming the subtle communication [149]. Binding studies of CLas-ZnuA2 by Surface Plasmon Resonance (SPR) revealed the low metal-binding affinity. Based on the structural and biophysical studies, it was hypothesized that CLas-ZnuA2 evolved to bind Mn^2+^ and reversibly Zn^2+^, which allowed Mn^2+^ transport and the avoidance of Zn toxicity in *C*Las [148]. The proteome analysis showed that the heavy metal permease and cation (Co/Zn/Cd) exporter system is present in *C*Las, but the specific metal uptake SBP for the heavy metal permease is absent. The crystal structure of CLas-ZnuA2 in Cd^2+^-bound form and binding studies by surface plasmon resonance (SPR) led to the hypothesis that CLas-ZnuA2 might be involved in sequestering (heavy metal) and transport of various divalent metals [146].

##### Amino Acid Uptake System

The proteome analysis of *C*Las showed the presence of two periplasmic amino-acid-binding proteins, namely cationic-amino-acid-binding (GenBank accession no. ACT56643) and putative cystine-binding proteins (GenBank accession no. ACT56645), which are components of the ABC transporter. The relative expression studies showed overexpression of cationic-amino-acid-binding protein in psyllid and cystine-binding protein in planta [135]. The sequence analysis of the cystine-binding protein of *C*Las (CLasTcyA) showed similarity with the periplasmic cystine-binding protein of *Neisseria gonorrhoeae*. Crystal structure and binding studies of CLasTcyA by SPR and microscale thermophoresis (MST) showed a maximum affinity with cystine ligand. Detailed crystal structure analysis of CLasTcyA showed some unique features, like a relatively larger binding pocket, presence of an extended c-terminal loop stabilized by a disulfide bond, and involvement of residue from the hinge region in the stabilization of the ligand [145]. Biophysical characterization of mutant CLasTcyA confirmed this unique feature. Comparison of the open and closed state of CLasTcyA showed only ~18.4° domain movement in the presence of the ligand, but ~40–60° domain movement was observed in other amino-acid-binding proteins [144]. Restricted domain movement allowed the CLasTcyA to capture ligands of different sizes and control the affinity of the binding. Proteome analysis of *C*Las showed the presence of different permeases, such as cysteine permease, generalL-aminoacid, and branched-chain amino acid (Valine/Isoleucine/Leucine) permease. Expression studies showed a higher expression of these permeases in planta [135]. In *C*Las, only two specific periplasmic amino-acid-binding proteins are present: one is cationic amino acid, and another is CLasTcyA. Based on the unique feature of CLasTcyA and the presence of permease, it was hypothesized that CLasTcyA might be involved in the transport of cystine and other branched-chain amino acids [145]. Insilico structure-based inhibitor screening against CLasTcyA showed five compounds in the zinc database (ZINC000000211883, ZINC000004707227, ZINC000013843286, ZINC000018063863, and ZINC000100640093) that exhibit higher binding energy than the ligand molecule [146]. Further biophysical and biochemical characterization might confirm the strength of binding. Field studies might also confirm the efficacy of inhibitor compounds.

##### Sec-Translocase/Translocon (SecY/SecE/SecG)

In bacteria, translocase and translocon (complex membrane transporter system) are involved in the transport of extracytoplasmic proteins into and across the inner membrane. The bacterial translocase is made of the core heterotrimeric “SecY/SecE/SecG” transmembrane protein and a peripheral ATPase motor (SecA) [151,152]. In bacteria, several vital proteins that are required for bacterial growth are secreted through SecA [153,154,155]. The SecA protein is highly conserved in bacteria and is also related with the virulence mechanism. It isa well-known drug target for developing antimicrobial compounds. It was demonstrated that “Rose Bengal (RB)” and its analogs act as SecA inhibitors for *E. coli* and *B. subtilis.* The crystal structures of *E. coli* “SecA” are available in the protein database (PDB) [156]. Homology modeling of the SecA protein of *C*Las (CLas_SecA) was done by using the PDB structure of *E.coli* SecA (PDB ID: 2FSG) as a template. A predicted model was used for structure-based virtual screening and molecular docking to discover a novel inhibitor molecule against CLas_SecA. The ATPase activity in the presence of an inhibitor with seventeen compounds showed IC_50_ > 50% inhibition, while four compounds had more than 65% inhibition [157].

Transcription regulator

Bioinformatics studies showed only 11 genes of *C*Las (2% of the whole genome) encode 19 transcription regulators which regulate all gene expression [56,158]. In *C*Las, a single transcription factor controls the several genes, so targeting the one transcription factor could affect the function of multiple genes pleiotropically. These greatly affected the bacterial adaptation and survival inside the host [159]. *C*Las spends its life cycle in citrus and insect hosts; therefore, it must have the ability to modify the gene expression in a host-specific manner to adapt in two different environments [158]. Studies showed that when *C*Las change their host from psyllid to citrus plant, the expression of a gene associated with survival and pathogenicity are up-regulated in planta. Expression of some transcription factors was also up-regulated in planta (Table 5). The transcription factors PrbP and LdtR of *C*Las were biochemically characterized [160]. The factor PrbP is a transcription activator and a predicted RNA-polymerase-binding protein in *C*Las, which interacts with the β-subunit of RNA polymerase (RNAP) and the short specific recognition sequence on the promoter region [159]. Small molecule inhibitors were identified by molecular screening assays, and therapeutic efficacies were tested against *C*Las. It was observed that the identified molecule, tolfenamic acid, significantly decreases the transcriptional activity of *C*Las and inhibits infection in citrus seedlings [159]. Transcriptome and in silicostudies predicted that LdtRn (MarR family transcriptional regulator) controls the expression of ~180 genes (like cell motility, cell wall biogenesis, transcription, and energy production) in *C*Las [160]. Biochemical screening (thermal shift assay) was performed to identify small lead molecules that modify the LdtR activity, and the biological impact was examined in *Liberibactercrescens* and *S. meliloti*. The high-throughput screening of small molecule inhibitors against the *C*Las transcription regulator was performed using an in vivo synthetic system which was designed using the closely related model bacterium *S. meliloti* as a heterologous host. The identified compound ChemDiv C549-0604 specifically inhibits the activity of *C*Las transcription regulator VisN [161,162,163,164,165], which suggests that the small lead molecules that target transcription regulators of *C*Las can potentially mitigate HLB.

Hydrolase family enzyme

The proteome analysis reveals a lot of important and potential drug targets, including serine proteases and phosphatases, which are hydrolase family enzymes. The structural and biochemical studies of these hydrolase enzymes will help in revealing the ligand- and substrate-binding sites, which could be an area for potential inhibitor development to control *C*Las.

##### Phosphatase

The phosphatases represent a part of bacterial signaling pathways and act as a virulence factor by interacting with the host signaling system [166]. Proteome analysis showed the presence of one hypothetical protein (ACT57371) in *C*Las that showed similarity with the dual-specificity phosphatase (DSP, protein, serine/threonine, and tyrosine phosphatase) family having a phosphotyrosine protein phosphatases II fold. In *C*Las, the cognate protein Ser/Thr or Tyr kinase of CLas_DSP was absent. Therefore, it is hypothesized that CLas_DSP might be participating in the signaling pathways of infected citrus plant and mimics the plant immune responses [167]. The N-terminal signal peptide is present in CLas_DSP, suggesting that it may be secreted to take part in virulence. The haloacid-dehalogenase (HAD)-like enzymes comprise a large superfamily of phosphohydrolases present in all organisms. HAD-like enzyme showed non-specific phosphatase activity, having a Rossmanoid fold [168]. The *C*Las genome also encodes the HAD family enzyme, HAD family hydrolase CLasHAD (EXU77906) and phosphoserine phosphatase SerBCLas_PSP (ACT57433).

##### Serine Protease (Protease IV Transmembrane Protein)

The membrane-bound self-compartmentalized serine protease, signal peptide peptidase A (SppA), cleaves the remnant signal peptide after the complete transport of the secretary protein at the targeted site. Bacterial SppA has a serine/lysine catalyticdyad mechanism to cut the signal peptide [169]. A few serine protease inhibitors have been reported against *E. coli* SppA protein, such as chymostatin, leupeptin, antipain, and elastinal [170]. The *C*Las genome encodes a protease IV transmembrane protein (ACT57220) (CLas_SppA) with a periplasmic signal peptide, which is up-regulated in the plant system [135].

Although *Murraya* species are members of Rutaceae family, they have not shown symptoms after *C*Las infection. The actual reasons for no symptom expression by the *Murraya* species is unknown [171]. It has been observed that miraculin-like protein (MLP) was overexpressed during *C*Las infection [172]. MLP expresses constitutively in the seeds of *Murrayakoenigi,* which has been purified and characterized [173]. MLPs have been reported to possess protease-inhibitory and antifungal properties [174]. It has also been reported that MLPs are overexpressed during *C*Las infection [172]. The role of MLPs during *C*Las infection should be investigated by studying the possible inhibition of important serine proteases of *C*Las. Another protein extracted from the *Putranjiva roxburghii* plant alsoshowed strong trypsin-inhibitory activity [175]. Treating *C*Las-infected citrus with antimicrobial compounds targeting critical proteins (serine proteases and phosphatases) could be an additional management approach for HLB disease.

Antioxidants Protein

Peroxiredoxin (Prx) protein plays an important role in the regulation of peroxide and protects organisms from peroxide-mediated oxidative damage. Prx protein is considered as an important protein of the antioxidant defense system of aerobic organisms, as they are involved in the hydrogen peroxide (H_2_O_2_) signaling pathway. *C*Las also has an antioxidant defense system, and the genome could encode different antioxidant proteins that protect from the lethality of the reactive oxygen species (ROS). These include cysteine-based Prx containing key residue, like peroxidaticCys (CPSH), embedded within the conserved PXXX (T/S) XXC motif. This protein family can be divided into two types based on the presence of cysteine. In 1-cys Prx, only peroxidatic cysteine is present, and in 2-cys, both peroxidatic and resolving cysteine are present [176]. *C*Las contains both 1-cys (Gene Id: ACT56685) and 2-cys (Gene Id: ACT56784) Prxs. The *C*Las 1-cys Prx protein (CLas_BCP) has CPSH/sulfenic acid cysteine (C46) and lacks the resolving cysteine (CRSH), which plays a major role in the *C*Las oxidative defense system. The purified *C*Las_BCP catalyzes the detoxification of peroxides using non-physiological electron donor DTT with the various substrates [72]. The protein protects the cell from H_2_O_2_-mediated cell killing and shows antioxidant activity by scavenging the reactive oxygen species (ROS). The invitro DNA-binding studies confirmed that CLas_BCP protects the supercoiled DNA from oxidative damage. Biophysical analysis of CLas_BCP by Circular Dichroism (CD) experiments showed that it is a β-sheet-rich protein [72]. A detailed biochemical and structural analysis of these enzymes along with their reductase partner will enhance our understanding of the structure–function relationships in *C*Las.

Nucleotide biosynthesis

Nucleotides are the building blocks of DNA and RNA with biosynthetic, de novo, and salvage pathways. The salvage pathway recovers the nucleotides formed during the degradation of nucleic acid (DNA and RNA), and de novosynthesis would be the main pathway for the formation of nucleotides from simple precursor molecules [177].

##### Bifunctional Enzyme 5-Aminoimidazole-4-Carboxamide Ribonucleotide Formyl Transferase/Inosine Monophosphate Cyclohydrolase (ATIC)

De novo nucleotide biosynthesis is the main pathway required for the formation of inosine monophosphate (IMP) from simple precursor molecules. In higher organisms, the pathway comprises of 10 steps, but in most microorganisms, 11 biosynthetic steps are required. In de novo purine biosynthesis, ATIC enzyme (encoded by the purH gene) catalyzes the penultimate and final steps of biosynthesis [178]. ATIC is a bifunctional enzyme; biochemical characterization of ATIC demonstrated that two activities reside on separate domains [179]. The *C*Las genome could encode bifunctional phosphoribosyl aminoimidazole carboxamide formyl transferase/IMP cyclohydrolaseenzyme (ACT57137). ATIC of *C*Las (CLas_ATIC), which is 536 amino acids long (M_w_ 59.04 Kda), showed maximum identity with ATIC of *M. tuberculosis* (PDB ID: 3ZZM). It is an essential enzyme for the survival of rapidly growing pathogenic bacteria. Developing inhibitor molecules against ATIC would help to impair the protein function [180]. Thus, the determination of the three-dimensional structure of the CLas_ATIC enzyme might pave the way for the design of novel inhibitors to potentially mitigate the impact of HLB disease.

##### Inosine-5′-Monophosphate Dehydrogenase

Inosine Monophosphate Dehydrogenase (IMPDH), involved in the de novo synthesis of purine, catalyzes the conversion of Inosine Monophosphate (IMP) into xanthosine-5′-monophosphate. This is the rate-limiting step in de novo guanine synthesis. The IMPDH from *C*Las (CLas_IMPDH) is a 493-amino-acid-long protein; its crystal structure has been submitted to the Protein Data Bank. The CLas_IMPDH (PDB ID: 6KCF) could be used for virtual screening and docking studies for the development of potential lead inhibitor molecules.

Fatty acid biosynthesis

Fatty acid biosynthesis (FAS) is important in all living organisms, including bacteria. It is essential for viability and a validated target for the development of antimicrobial molecules. On the basis ofthe enzymes involved, fatty acid biosynthesis is classified into two different pathways: FASI (Type I) and FASII (Type II). FAS I is a multifunctional enzyme with multiple domains that is found in mammals and fungi [181]. In FASII, each step is catalyzed by monofunctional enzymes found in plant chloroplasts and bacteria. Enzymes involved in FASII were observed as highly specific in bacteria [182].

##### Enoyl-Acyl Carrier Protein Reductase I (FabI)

Type II fatty acid synthase and enoyl-acyl carrier protein reductase I (FabI) enzyme catalyzes the final step of bacterial fatty acid biosynthesis. FabI is a crucial enzyme in the completion of cycles, particularly in the elongation phase of fatty acid biosynthesis [183]. Crystal structures of several FabI from different organisms have been reported in the Protein Data Bank (PDB). The *C*Las genome encodes 30.8 kDa enoyl-acylcarrierprotein reductase [NADH] enzymes (CLas_FabI). CLas_FabI crystalized in apo (PDB ID: 4NK4) and in complex with NAD^+^ (PDB ID: 4NK5) showed the conformational change in the substrate binding loop. Inhibitor kinetics showed that isoniazid (INH) acts as a competitive inhibitor with respect to NADH substrate and an uncompetitive inhibitor with crotonoyl-CoA [184]. Therefore, the structure of CLas_FabI would be used for virtual screening and molecular docking for the identification of potential lead inhibitor molecules.

##### β-Hydroxyacyl-acyl Carrier Protein Dehydratase (FabZ)

In the Type II fatty acid biosynthesis pathway, the FabZ enzyme is involved in dehydration of β-hydroxyacyl-ACP to trans-2-acyl-ACP. 3-hydroxyacyl-ACP dehydratase (FabZ) (WP_015452391) of *C*Las (CLas_FabZ) has been shown to be an 18kDa protein, and a sizeexclusion study confirmed a hexameric form in solution. The crystal structure (PDB ID: 4ZW0) analysis of CLas_FabZ showed similarity with other reported structures of FabZ in a different organism [185]. The crystal structure of CLas_FabZ provides important insights for the development of antibacterial molecules.

Amino Acid Biosynthesis

In *C*Las, enzymes involved in the conversion of phenylpyruvate to phenylalanine and asparate to lysine were identified. Furthermore, enzymes involved in the biosynthesis of amino acids from metabolic intermediates have been identified. The enzyme for the biosynthesis of amino acids (tyrosine, leucine, isoleucine, tryptophan, and valine) from metabolic intermediates is absent [56]. In the diaminopimelate pathway of lysine biosynthesis, the enzyme dihydrodipicolinate synthase (DHDPS) was involved. DHDPS catalyzes the condensation of pyruvate with L-aspartate beta-semialdehyde. Recently, the crystal structure of DHDPS bound with pyruvate (PDB Id: 7LOY) and allosteric inhibitor (S)-lysine (PDB Id: 7LVL) from *Candidatus* Liberibacter solanacearum (CLso_DHDPS) has been submitted to the Protein Data Bank [186]. Dihydrodipicolinate synthase of *C*Las (CLas_DHDPS) has 80.82% sequence identity with CLso_DHDPS. This structural information of CLso_DHDPS provided the basis for insilico inhibitor studies against CLas_DHDPS.

To date, only five proteins (two transporters and three enzymes) from *C*Las have been crystalized in apo and complex form, and 17 structure coordinates are available in the Protein Data Bank (Table 6). Structural studies of important transporters and enzymes from *C*Las revealed some unique features, which are different from other reported structures of a similar protein in different bacteria. These unique features could be targets for the virtual screening of potential inhibitor molecular candidates available in database libraries.

#### 4.1.2. HLB management with Antimicrobial Chemicals

The effectiveness of a variety of broad-spectrum antibiotics has been reported on HLB-infected trees under greenhouse as well as field conditions. Foliar application of effective antibiotics has been recommended to alleviate the *C*Las population and greening management in citrus [120,188]. The application of antibiotics like achromycin andledermycin on citrus branches has helped to suppress the symptoms of HLB [189]. The antibiotics oxytetracycline and penicillin were found effective against citrus greening to reduce disease severity [190]. Zhang et al. (2010, 2011, and 2012) have demonstrated that the combination of penicillin and streptomycin is effective for suppressing the *C*Las titer in HLB-affected citrus [90,191,192]. Other antibiotics found effective in suppressing *C*Las populations include ampicillin, carbenicillin, penicillin, cefalexin, rifampicin, and sulfadimethoxine [120]. Oxytetracycline and streptomycin received emergency approval to be used in citrus groves in Florida for foliar use by the United States Environmental Protection Agency [193], and the approval for streptomycin was extended for 7 more years in 2021. However, the effectiveness of these treatments for suppressing HLB is still an ongoing debate [188]. Studies indicate that due to the nature of the cuticle layer on citrus leaves, the antibiotics applied foliarly have very limited uptake, therefore not reaching target sites where the *C*Las reside [63,194]. Therefore, a delivery system is necessary for antimicrobial chemical components to get to the phloem, a targeted site. A proofofconcept was conducted by Killiny et al. (2020) to demonstrate if the cuticle layer is the main barrier for the uptake of oxytetracycline [195]. Some parts of the cuticle layer of citrus leaves were removed using a high-power laser followed by foliar application of oxytetracycline [195]. A significant amount of absorption of the antibiotic was observed, resulting in a decrease in *C*Las titer in comparison to the control. This study suggested that the cuticle layer was the first and major barrier for oxytetracycline bioavailability for killing *C*Las. Authors also tested combining oxytetracycline with adjuvants commonly used in the agricultural industry to see if it would facilitate oxytetracycline absorption [195], but mixing with the adjuvant had minimal impact on the uptake of oxytetracycline. The trunk injection method offers tools to directly administer agrochemicals to the plant vascular system. Use of this technique would regulate the amount and specificity of agrochemical introduced into the plant vascular system based on the disease, age, and size of trees. Previous reports have revealed that trunk injection method is an effective way to overcome the limited absorption of the antibiotics by the citrus leaves [196,197]. Al-Rimawi et al. (2019) demonstrated that oxytetracycline and streptomycin antibiotics were still detectable even after thirty-five days of treatment applied via soil-drench or trunk-injection methods [197]. The residual activity could provide long-term protection and decrease frequency of treatment application. The possibility of bacteria developing resistance against antibiotics cannot be ruled out. Additionally, the possibility of antibiotic contamination of the fruits must be assessed carefully. Therefore, to successfully manage HLB in the field, antibiotic alternatives should be investigated.

Brassinosteroids are a family of plant steroidal compounds that play an essential role in plant growth, development, and stress tolerance. Studies have revealed that brassinosteroids can be used as a possible control strategy for HLB. The foliar spray of epibrassinolide has been recommended for *C*Las-infected citrus plants.It was observed that *C*Las titers were reduced by treatment with epibrassinolide under both greenhouse and field conditions [198]. The reduction in pathogen titer was the consequence of the induction in the defense-related genes in the leaves of the citrus plant. The role of brassinosteroids has also been observed in the induction of the H_2_O_2_ in maize leaves [198].

SA and JA play a major role in the plant defense mechanism that induces the expression of the pathogenesis-related (PR) proteins in response to a pathogen attack. SA mediates the phenylpropanoid pathway to defend against pathogens, insect pests, and abiotic stresses. JA mediates the octadecanoid pathway as the defense against insect pests and pathogens [199]. A common defense system which is activated by both JA and SA was proposed in rice [200]. SA blocks H_2_O_2_ inhibitors or scavengers, resulting in elevated levels of H_2_O_2_ and the activation of defense-related genes. It was also demonstrated that SA and its synthetic inducer 2, 6-dichloroisonicotinic acid inhibit ascorbate peroxidase (APX), which is also an involved defense mechanism. Other plant defenseinducers were able to induce defense mechanisms against a different disease, such as β-aminobutyric acid, 2,1,3-benzothiadiazole, ascorbic acid, 2-deoxy-D-glucose, and 2,6-dichloroisonicotinic acid [201].

Australian finger lime (*Microcitrus australiasica*), an inimitable citrus fruit that grows in Australia’s rainforests, has recently been reported as immune to HLB [202,203]. The comparative analysis among the HLB-tolerant and HLB-sensitive citrus cultivars demonstrated stable antimicrobial peptides (SAMPs) responsible for the tolerance of finger lime against HLB [204]. The peptide made by finger limes, SAMPs are involved in enhancing host immunity and inhibiting the *C*Las multiplication in HLB-positive trees. It has been reported that heat stablized SAMP (130) applied through foliar spray moved systemically in the vascular system, making them more suitable for field applications than antibiotics [204].

##### Engineered Nanomaterials (ENMs) for HLB Management

Nanotechnology is one of the fastest-growing scientific fields over the last few decades. The incorporation of nanotechnology-enabled tools and materials in the agricultural industry is inevitable. In a broad definition, ENMs (Engineered Nanomaterials) are substances with at least one of the dimensions between 1 nm to 100 nm [205,206]. ENMs are expected to improve the overall health of plants in accordance with sustainable agricultural practices [207]. Until now, ENMs for agricultural applications were studied as sensors [208], fertilizers [209], andurea-coating [210] and soil-conditioning agents [211,212]. For disease management, they can be developed as nano-carriers to facilitate the transportation of active ingredients (A.I.) to the diseased sites or used directly, thanks to their intrinsic biocidal properties [213]. ENMs are reported to have higher antimicrobial efficacy compared to corresponding bulk counterparts due to their higher surface-to-volume ratio; therefore, it is reasonable to expect that they can work more efficiently at lower concentrations or require smaller treatment frequencies compared to corresponding bulk counterparts [209,214].

In order to adapt nanotechnology-enabled materials for the management of HLB, it should reach phloem tissue at a concentration that is suitable for inhibiting *C*Las growth (i.e., above the minimum inhibitory concentration). ENMs applied via surface application methods have barriers to overcome before reaching the plant vascular system [215] (Figure 8). Foliarly applied ENMs have two possible paths of entry: stomatal or cuticular. Soil-drench-applied ENMs need to be absorbed via the casparian strip [216]. It has been observed that there is sizeexclusion for uptake and movement of ENMs into the phloem vascular tissue of citrus trees after foliar application [217]. Results indicated that 5.4 nm and below was optimal for nanomaterials to pass through epidermis tissues and mobilize into the phloem channels [217]. However, previous studies indicate that surface coating [218] and shape [219] of ENMs were other factors affecting the uptake and translocation in the plant vascular system. To understand how the surface chemistry and application method affects the uptake and movement ofsilver nanoparticles (AgNPs) in citrus trees, Suet al.compared the uptake and mobility of AgNPs coated with citrate (28.7 +/− 11.0 nm), polyvinylpyrrolidone (17.9 +/− 7.5 nm) (PVP) polymer, and gum Arabic (9.2 +/− 4.2 nm) (gA) in 2.5-year-old mandarin trees [220]. Colloidal solutions were applied using foliar spray (drop casting), petiole feeding, root-drench or trunk injection methods [220]. AgNPs applied via all four application methods were absorbed by the plant tissues, while surface coating was found to be more critical for systemic movement of the NPs in the plant vascular system. PVP- and gA-coated AgNPs distributed fully via both xylem and phloem channels, while citrate-coated AgNPs aggregated in the trunk due to the high salinity of the sap [220].

ENMs can have intrinsic biocidal properties due to retaining antimicrobial metals such as Ag, Zn, and Cu. Until now, the antimicrobial efficacy of AgNPs was reported against different plant pathogens [221]. The use of AgNPs in agricultural practices is promising because studies have demonstrated treatments that can be used for pathogens that already developed resistance to Cu-based antimicrobials [222]. Release of antimicrobial Ag^+^ ions causing cell membrane rupture is believed to be the major path of invitro biocidal efficacy of AgNPs [223]. Stephano-Hornedo et al. reported the effectiveness of AgNPs suppressing *C*Las in Mexican limes in a field study [224]. AgNPs (35 +/− 7 nm mean particle size, zeta potential −14 mv) were applied via either trunk injection or foliar sprinkle methods, and changes in *C*Las titer were quantified to determine the effectiveness of the treatments. The study highlighted that both application methods are effective for the management of HLB. Up to 90% reduction in bacterial titer was measured after foliar application, while the trunk injection method reached 80% bacterial titer decrease at 12.5 times below the concentration [224]. One must note that Ag in the ionic form is toxic to other bio-organisms in the soil and water as well [225,226,227]. Therefore, more in-depth nanotoxicology studies are required to analyze long term benefits and cost-effectiveness.

Zn and Cu are two micronutrient-based metals that possess toxicity against plant pathogens yet are beneficial to a plant’s overall health by participating in biomolecule synthesis, boosting the plant’s defense mechanism [209]. Cu as an antimicrobial agent for agricultural applications has been recognized by regulatory agencies worldwide. Just in Florida, ~530 tons of Cu hydroxide per year were applied for disease management between 2007 and 2009 [228]. However, conventional Cu products are ineffective at managing HLB due to a lack of systemic uptake and mobility [229]. It is fair to assume that formulations retaining nano-Cu as the A.I. can overcome the size limitation of commercial products. However, to the best of our knowledge, there is no nano-Cu formulation available for controlling *C*Las. Nanomaterials have higher solubility in aqueous media compared to bulk forms [230]. This could be a major limitation for developing a non-phytotoxic systemic Cu bactericide formulation for killing *C*Las.

It has been reported that HLB infection causes micronutrient Zn imbalance within the plant system [231]. Therefore, it is expected that Zn-containing bactericide formulations would exhibit additional nutritional benefits. Previous reports demonstrated that chelating Zn metal with organic lipophilic ligands can enhance its antimicrobial efficacy [232,233] and improve their uptake by citrus leaves compared to bulk ZnO [234]. A multifunctional agrochemical formulation (MS3T) containing a Zn chelate, quarternary ammonium compound (Quat), and clay have been shown to be effective against citrus canker citrus melanose and scab in field conditions [235]. MS3T enabled surface protection by creating a clay-supported filmbarrier between leaves and disease-bearing insects and released antimicrobial Quat and chelated form of zinc [235]. Field trial results demonstrated that the formulation is as effective as commercial Cu standards in controlling disease. Importantly, the MS3T formulation improved fruit yield and quality, suggesting the nutritional benefits of Zn. Zn concentration in the leaves of the treated trees increased significantly when compared to untreated leaves, suggesting Zn uptake. In a separate study, water-soluble Zn salt (Zn sulfate) and Zineb (a polymer dithiocarbamate complex of Zn) was applied to HLB-bearing trees, and disease progression was monitored via *C*Las titer [236]. It is interesting to note that even though Zn levels increased in the leaves after the application, HLB symptoms were not mitigated. In fact, increase in *C*Las titer was observed due to beneficial effects of low concentrations of Zn on bacteria growth [236]. This suggests that the Zn level in phloem tissue is critical for mitigating HLB symptoms.

Until now, ZnO-based ENMs have been studied more extensively as a fertilizer for agricultural practices. Studies on varieties of plants suggest that ZnO-based ENMs can improve plant productivity [237], root growth [238], photosynthesis rate [239], seed germination rate [240], and plant biomass [241]. Biocidal properties of ZnO-based ENMs can be attributed to the release of antimicrobial Zn and the generation of reactive oxygen species (ROS) [242]. ROS are generated when ENM takes part in the catalytic activity. A ZnO nanoparticle with surface defects serves as the catalytic sites and is responsible for producing ROS such as peroxides, superoxides, and hydroxides [243]. In general, ENMs’ catalytic activity is triggered bythe absorption of a photon, which is unique to nanoparticles, but this property is minimal/absent in their larger counterparts [244]. Studies indicate that the ROS interfere with several important cellular processes for viability at the molecular level, causing cell death [245].

To combat against HLB, a formulation of ultra-small (<10 nm) nano-ZnO (Zinkicide^®^) was developed [246,247]. The superior antimicrobial property of Zinkicide has been correlated with the surface-defect-related catalytic activity of the nano-ZnO [247]. Surface defects originate at the atomic level due to the presence of Zn or O vacancies [247]. Naranjo et al. (2020) studied the mechanism of action of Zinkicide on *Liberibacter crescens* in batch cultures and in microfluidic chambers as the model system mimicking the plant vascular system [248]. Much-improved in vitro antimicrobial efficacy was reported for Zinkicide in comparison to bulk ZnO (300–900 nm). It was observed that Zinkicide particles released higher levels of Zn ions that contributed to intra-cellular ROS generation followed by lipid peroxidation and even cell membrane disruption. Zinkicide demonstrated better performance than bulk ZnO in inhibiting the bacterial biofilm of *Liberibacter crescens* between the 2.5 and 10 ppm Zn level. This study was carried out using microfluidic chamber assays mimicking phloem vascular tissue [248]. This study also demonstrated that nano-ZnO had higher mobility and bactericidal efficacy in sink reservoirs of the microfluidic chamber; therefore, it was more effective at clearing the biofilm compared to bulk ZnO. Soliman et al. (2022) reported ZnO/ZnO_2_ (2.7 nm core, 0.4 nm shell as determined via SANS measurements) nanotherapeutic formulations against HLB and canker-bearing citrus trees [249]. The authors discovered a significant decrease in canker lesions compared to Firewall^®^ control and in the HLB disease rating compared to Nordox^®^ 30/30 control. It must be also noted that the ZnO/ZnO_2_ formulation caused a percentage increase in large fruit size yield compared to control, which is desired for the fresh fruit market [249]. Graham et al. (2016) reported the efficacy of Zinkicidefor controlling the citrus canker of grapefruit, another important disease [250]. Zinkicide-treated trees showed a low level of disease incidence (9.2%) in comparison to untreated controls (63%) and Cu oxide industry controls (21%). It is also interesting to note that the application rate of ZnO in the field was half the rate of total metal concentration compared to the Cu commercial control. The effectiveness of nano-ZnO formulation combined with the 2S-albumin protein on HLB-infected seedlings was reported [106]. The formulation consisted of nano-ZnO (~4 nm) capped with an organic coating that stabilized the 2S-albumin protein at neutral pH via intermolecular interactions. In this study, 3-year-old mosambi trees were injected with a colloidal solution of nano-ZnO, 2S-albumin, and nano-ZnO plus albumin, and *C*Las titer was quantified for 120 days. A significant decrease in titer at 28-fold with 2S-albumin, 26-fold with nano-ZnO, and 34-fold for nano-ZnO plus albumin formulations were observed [106,251]. Zn nitrate administered alone did not show any decrease in titer, consistent with previous reports [236]. As an alternative to the conventional trunk injection method for the administration of ENMs, Kundu et al. (2019) developed micromilled microneedles for the precise delivery of agrochemicals, including antimicrobial nano-ZnO (Zinkicide) solution [252]. Micro-punctures were created on the bark of citrus seedlings, and the Zinkicide-solution-soaked pad was wrapped around the treated area. A significant increase in metallic Zn was observed on the stem and leaves of seedlings treated with microneedles compared to untreated seedlings [252].

An ENM platform with no intrinsic antimicrobial properties can be used as a delivery system to improve the effectiveness of conventional antimicrobials. Polymeric EMNs consisting of hydrophobic and hydrophilic moieties can encapsulate lipophilic antimicrobial active ingredients and release them at the target site. A bio-degradable polysuccinimide (PSI) polymer nano-delivery system (average particle size 20.6 +/− 0.6 nm, zeta potential −28.5 mV +/− 0.7 mV) was reported [253]. Grapefruit cell suspensions were incubated with PSI nanoparticles loaded with a model fluorescent compound, and the distribution of the fluorescent signal in the cytoplasm and nucleus was observed after 2 h [253]. Another way that ENMs can be used to improve the effectiveness of current antimicrobials is by improving their rainfastness. Kah et al. reported that nano-formulations applied on citrus leaves have better rainfastness compared to bulk counterparts due to higher surface area interacting with the leaf surface [254]. Maxwell et al. developed a novel nano-ZnS (particle size ~3.5 nm) adjuvant to improve the rainfastness of streptomycin sulfate. In this study, the rainfastness property of a mixture of streptomycin and N-acetyl-cysteine-coated nano-ZnS was compared with streptomycin sulfate alone and a commercial product, Firewall^TM^ 50W [255]. After two simulated rainfalls, about 50% of the streptomycin was found to be adhering to the leaf surface when delivered through nano-ZnS, while 70% of the materials washed away from controls [255].

Even though the use of ENMs could be a promising strategy for HLB management in the field, to the best of our knowledge, there is no commercial product available to citrus growers yet. A formulation retaining ENMs listed as the antimicrobial active ingredient will have to go through registration and approval processes by regulatory agencies. Due to environmental and human exposure, ENMs are under high public and regulatory scrutiny. However, there are still no clear guidelines for biocidal applications of ENMs, and a universally accepted legal framework is needed urgently. Lawmakers need to adopt sensible analytical techniques to identify and prove the existence of ENMs and provide guidelines on the toxicity tests required [256]. Taking into consideration that one solution may not fit for all formulations, regulatory agencies must continue to work alongside scientists to adapt ENM technology. ENM products aiding in sustainable agricultural practices are critical for public and regulatory agencies’ acceptance and adaptation of this new technology [257]. For example, the use of chemicals designated Generally Recognized As Safe (GRAS) could possibly facilitate the process for approval [258]. Furthermore, environmentally friendly synthesis methods requiring a minimal amount of energy, number of synthesis steps, and purification are ideal while keeping the cost low enough to be able to compete with existing products [211,259]. Lastly, more in-depth residue studies are needed to understand the fate and interactions of ENMs with soil and fruits before commercialization.

#### 4.1.3. Thermotherapy and Cryotherapy

Thermotherapy has been used for decades in plants to control or kill the different micro-organisms as a good alternative to chemical-based pathogen control. This approach utilizes eitherdry or wet heat. Optimized treatment conditions include treatment duration and frequency. Thermotherapy is shown to be effective in reducing *C*Las titer without any significant damage to the plant. There are different ways of applying heat as treatment for plants, such as vapor heat treatment, soaking in hot water, and hot air. Researchers and growers have implemented the thermotherapy concept in the field on a small as well as large scale by using translucent plastic coverings to elevate tree temperatures. It is challenging to apply uniform heat to the tree canopy and the root system. An increase in canopy temperature beyond the optimum level may cause thermal injury to the tender top portion of the tree but may not be sufficient for the root system. A root system that escapes thermotherapy treatment may be responsible for the re-infection of *C*Las to the entire canopy. Application of the thermal treatment (40–42 °C for seven to ten days) under a controlled greenhouse condition eliminated *C*Las completely for as long as two years [258]. Fan et al. (2016) assessed the efficacy of short, repetitive thermotherapy of 4 h over a period of three weeks against *C*Las and found a significant reduction in titers in both 45 °C and 48 °C treatments in infected seedlings [260]. A novel method consisting of a combination of thermotherapy (45 °C) with sulfathiazole and sulfadimethoxine sodium forcontrol of HLB has been attempted [261]. The different thermotherapy treatments concluded that this concept could not completely eradicate the *C*Las under the field condition, although the treated trees expressed vigorous growth. It has been reported that many genes involved in plant–bacterium interactions were up-regulated after thermotherapy [262]. Nowadays, a mobile thermotherapy delivery system (mobile heat treatment system, MHTS) has been developed and used for in-field treatment of HLB-infected trees [263]. This system consists of different components required for heat generation and application of heat to the tree, including a tree canopy cover (hood), water tank, pump, generator, steamer, pressure washer, and a water softener. The system covers the canopy of the HLB-infected trees and injects the steam and hot water through nozzles inside of the hood to increase the temperature of the canopy. MHTS was also evaluated in the field by using a bio-based sensor (a surrogate bacterium, Klebsiella oxytoca). It was reported that the system showed good killing efficiency (3.35 log reductions in colony-forming units to the complete elimination of the bacteria) at a raised maximum temperature of 54 °C for 250 s [264]. Therefore, thermotherapy treatment could be one of the effective control strategies in an integrated management of HLB.

Cryotherapy of shoot tips is another popular technique used for the eradication of virus and virus-like pathogens from explants [265]. The frequency of obtaining pathogen-free plants is higher with the cryotherapy of shoot tips than with shoot tip culture. In cryotherapy, plant pathogens are eliminated from shoot tips by exposing them briefly to liquid nitrogen (−196 °C). After a brief cryo-treatment, shoot tips proceed for regeneration of healthy shoots to obtain pathogen-free plants. Thermotherapy followed by cryotherapy of meristems tissue can be used for efficient removal of virus and virus-like pathogen [266], as shown in the schematic representation (Figure 9). Ding et al. (2008) successfully used a vitrification-cryopreservation for removal of *C*Las fromthe infected plants of several citrus species [267]. They observed that almost all regenerated plants were HLB-free after combined cryotherapy and thermotherapy [267]. This procedure could be carried out in a tissue culture laboratory equipped with basic infrastructure for producing pathogen-free plants.

## 5. Management Strategies at Host Level

### 5.1. Activation of Plant Immune System

In response to pathogen attack, plants induce a series of defense responses, including the production of reactive oxygen species (ROS), synthesis of PR proteins, cell wall modifications, production of phytoalexins, and a hypersensitive response. The plant cell membrane consists of pattern recognition receptors (PRRs), which act as cellular ‘antenna’ to sense extracellular signals and allow plants to detect a wide range of danger signals, including pathogen-, microbe-, and virus-associated molecular patterns (PAMPs, MAMPs, and VAMPs). PRRs of the plant cells are involved in self-defense against attackers by triggering innate immune responses [268]. The citrus plant exposure with the pathogen (PRR-PAMPs interaction) induces the transient production of different molecules involved in the plant defense system such asreactive oxygen species, the activation of the mitogen-activated protein kinase (MAPK) cascade, nitric oxide burst, ethylene production, calcium influx, callose deposition at the cell wall, and expression of defense-related genes called PAMP-triggered immunity (PTI) [269]. To suppress the PTI of the host, pathogens have evolved a virulence mechanism that enables them to directly inject effectors into the host cell, which avoids PRR recognition. As a countermeasure against the effector molecules, the plant induces another defense mechanism called effector-triggered immunity (ETI). ETI is involved in the recognition of pathogen effector molecules by producing disease resistance (R) proteins and causes a hypersensitive response to kill the infected cells.

*C*Las is a resilient pathogen capable of suppressing the immunity triggered by the pathogen-associated molecular pattern (PAMP) of the citrus host. On the other side, *C*Las lacks type II plant-cell-wall-degrading enzymes, which are needed for the induction of the plant defense system. It is conjectured that the origin of the *C*Las is an insect/animal [270]. This may be the reason why the plant could not evolve to develop resistance/immunity against *C*Las. Therefore, the citrus plant fails to defend against HLB [271]. The pathogen–host interaction affects the expression of various gene patterns. The infection causes down-regulation of the heat shock protein genes, genes involved in the synthesis of cytokinins and gibberellins, while genes involved in the light reactions of photosynthesis, ATP synthesis, and ethylene pathways are up-regulated [272]. It also affects the various pathways involved in cell signaling, hormone synthesis, sucrose-starch metabolism, source-sink communication in phloem tissue, and the response through both SA and JA pathways [272]. In addition to local defense, plants can activate systemic acquired resistance (SAR). This is a broad-spectrum mechanism which involves the synthesis of chemical mobile signals in tissues that have been locally infected by pathogens. These signals travel throughout the phloem tissues, triggering plant defense responses in the uninfected portions of the plant. Thus, the activation of the plants’ own defense systems, by engineering the plants to synthesize R-proteins or defense-related signals, is another approach used for the improvement of citrus. Plants also employ antimicrobial peptides (AMPs) as a defense mechanism. These cationic molecules can interact with the negatively charged pathogen membranes by changing their electrochemical potential and altering the membrane permeability, which eventually leads to cell death. Because of their antimicrobial activity, broad spectrum, and low cytotoxicity, AMPs could potentially replace traditional antibiotics. As there is no immune cultivar available, it is therefore difficult to discover some plausible clue about greening tolerance. It was observed that grape fruit showed some tolerance against citrus greening [273]. Transcriptome analysis of US-897 (*C. reticulata* Blanco) cultivar identified two genes, 2-oxoglutarate and Fe (II)-dependant oxygenase, as responsible for the disease tolerance [274,275].

### 5.2. Transgenic Approach

To mitigate HLB, there is a need to trigger the citrus plant’s immunity against the *C*Las pathogen. In the past, conventional breeding methods have been used to improve citrus cultivars to develop new varieties. This traditional breeding method has some limitations due to the woody nature of the plant, juvenility, incompatibility, heterozygosity, polyembryony, parthenocarpy, and male or female sterility. Additionally, the traditional breeding of citrus cultivars was time-consuming and restricted to the traits related to fruit quality, like fruit ripening time, seed number, and flesh color [276]. Therefore, genetic engineering is an attractive approach for faster development of disease-resistant citrus plants and could provide an advantage for the combined management of citrus diseases like HLB. The immunity of the citrus plants can be triggered by expressing different genes involved in the plant defense system or suppressing the expression of some important *C*Las pathogenicity genes. The various strategies that have been used to enhance resistance/tolerance in citrus against HLB include overexpression of endogenous or exogenous antimicrobial protein and exploration of host–pathogeninteraction pathway [277]. The Environmental Protection Agency in the United States has approved the field testing of a transgenic citrus cultivar expressing spinach defensin genes against HLB [278]. The expression of the NPR1 gene of *Arabidopsis* spp. in citrus enhanced resistance against HLB, which induces the expression of several native genes involved in the plant defense signaling pathways by acting as a regulator of the transcription factor. The transcription factor induces the expression of the PR (pathogenesis-related) gene and ultimately mediates the SA-induced SAR pathways [279] (Figure 10). According to Wang et al. (2016), HLB-tolerant ‘Jackson’-grapefruit-like hybrid trees expressed 619 genes differently than susceptible cultivars (Marsh tree) [280]. It has also been reported that the cationic lytic peptide cecropin B from the Chinese tasar moth (*Antheraea pernyi*) is effective against *C*Las when expressed in the phloem tissues of citrus using a GRP1.8 promoter [275].The transgenic citrus expressing natural lytic peptide cecropin B has decreased their susceptibility to greening and showed strong activity against *C*Las [275]. Ramadugu et al. (2016) have conducted a six-year field trial to identify resistance against citrus greening among citrus-related subfamilies [281]. The high level of tolerance was observed in various non-citrus genera, e.g., *Eremocitrus* and *Microcitrus*, which are sexually compatible with sweet orange and grapefruit. These Australian citrus relative genera could be useful for improvingthe citrus genome against HLB through breeding trials or genetic engineering [281].

The flagellin protein of bacteria is also recognized as PAMP by the R-gene flagellin receptor 2 (FLS2). It is observed that the flagellin peptide (flg22Xcc) of *Xanthomonas campestris* pv. *citri* (Xcc) induces ROS and the expression of a defense-responsive gene in resistant citrus genotypes ‘Sun Chu Sha’ mandarin and ‘Nagami’ kumquat [282]. The overexpression of the FLS2 protein causes the reduction in susceptibility to Xcc. The FLS2 protein is also present in the sweet orange citrus but has very low homology to the known functional FLS2 isoforms, which may be the reason for the lack of a defense response [283]. The grapefruit citrus plants expressing FLS2-1 and FLS2-2 have increased the expression of the defense-related genes WRKY22, GST1, and EDS1 [283]. These resistance genes could be used to generate cisgenic lines to enhance PTI in the citrus against the bacterial diseases like citrus canker and HLB. There are various R-genes that can be used as potential targets in citrus improvement programs, such as the NDR1 gene (a positive regulator of salicylic acid accumulation), *Xa21* from rice (already used in the citrus cultivar Hamlin, Natal, and Pera, against the Xcc)*,* and the disease susceptibility gene (*CsLOB1* gene). Currently, research has emphasized genome-editing tools like CRISPR/Cas9 and single-guide RNA (sgRNA) for crop improvement. CRISPR/Cas9 is a specific and straightforward method used to alter the genome of crops to generate disease-resistant/tolerant plants, and it might be the most accepted method for crop improvement in the future [284,285]. Furthermore, genome-editing technology has already been used in citrus: a canker disease susceptibility gene *CsLOB1* has been modified to generate a canker-resistant plant [286,287]. These emerging and efficient genome-editing tools could be used in the triangular approach for complete eradication of HLB in the near future.

In addition to structure-based inhibitor studies, the foreign resistance (antimicrobial and insecticidal) genes derived from other species can be introduced into the citrus host by genetic engineering [288]. Plant seed storage protein performs vital roles in plant survival, acting as molecular reserves for plant growth, maintenance, and defense mechanism by virtue of their antimicrobial and insecticidal properties [289]. Antimicrobial molecules present in plants and many animal species involved in the defense mechanism are known as AMPs. The plant serine proteinase inhibitors play an important role in plant defense against pathogens and pests [290]. For instance, the genes encoding potent protease inhibitor and AMPs might be used as sources of the transgene for the development of citrus resistance [291]. The genome sequencing and transcriptome studies demonstrate the structural diversity and expression of constitutive disease resistance (CDR) genes in HLB-tolerant trifoliate orange (*Poncirus trifoliata*) and its hybrid plant. This study validated the potential role of CDR genes in HLB development and provided insight into the genetic manipulation of the citrus plant [292]. Different potential antimicrobial peptides which have been studied against different pathogens include Cercopin B, Attacins, Thionin, D2A21, Dermaseptin, Sarcotoxin IA, and Linalool. The transgenic sweet orange plants expressing phloem-specific antimicrobial geneattacinA have been evaluated against HLB [293]. The thioninis is a family of AMPs which attack the bacterial cell membrane, causing cellular leakage and death. Transgenic Carrizo rootstocks expressing a thionin gene exhibited reduced canker symptoms with a concomitant decrease in bacterial growth. When inoculated with HLB-infected budwood, these transgenic plants had lower *C*Las titers in comparison to the controls. These findings indicate that the modified thionin might be a helpful AMP in the fight against citrus bacterial diseases [283]. The use of two or more transgenes (with different defense mechanisms) for the development of transgenic citrus plants would prevent the emergence of a mutant strain of pathogen. It is possible to overcome the HLB crisis with the development of transgenic citrus crops that have multiple defense genes (antimicrobial and insecticidal).

#### 5.2.1. Emerging Potential Genes as Transgene

##### 2S Albumin

Based on solubility, plant seed storage proteins have been classified into different groups, such as albumin (water), globulins (saline), prolamins (alcohol/water), and glutelins (alkali). Furthermore, these proteins are classified based on their sedimentation coefficients [294]. The term 2S albumin is defined based on their water solubility and sedimentation coefficient [295]. 2S albumins are small heterodimeric proteins having two chains (~3–4 kDa for small and 6–9 kDa for big chains), and both chains are stabilized by inter- and intra-chaindisulfide bridges. 2S albumin from *Brassica napus* (napins) is a basic protein (pI ~11) having a broad range of antibacterial activity and antifungal activity against *Fusarium oxysporum*, *Fusarium culmorum, Botrytis cinerea*, and *Alternaria brassicola*. It is believed that the antimicrobial activity of napin is due to the presence of a high proportion of positivelycharged amino acids, calmodulin antagonist, and trypsin-inhibitory activity [296]. 2S albumins from *Brassica oleracea* (kohlrabi), *Sinapis arvensis* (charlock), and *Brassica nigra* (black mustard) showed trypsin- and subtilisin-inhibitory activity, and 2S albumin from *Sinapis arvensis* also showed α-chymotrypsin-inhibitory activity [297,298,299]. The first crystal structure of 2S albumin from *Moringa oleifera* (*Mo*-CBP3-1) has been determined. It is made up of two proteolytically processed α-helical chains, stabilized by four disulfide bridges. *Mo*-CBP3-1 is thermostable (melting temperature ~98 °C) and pH-resistant. The presence of a polyglutamine motif and surface arginines has supported its antifungal and antibacterial activities [300]. The 16 kDa 2S albumin (heterodimer of ~11 kDa and ~5 kDa) from *Wrightia tinctoria* (WTA) has Dnase and antibacterial activity against the human pathogen *Morexella catarrhalis* [301]. The 2S albumin from *Putranjiva roxburghii* is thermally stable and exhibits antifungal, DNase, RNase, and in vitro translational inhibitory activities [302]. Biochemical characterization of pumpkin (*Cucurbita maxima*) 2S albumin showed antifungal, DNase, RNase, and cell-free translationalinhibitory and anticancer activities [303]. The plant efficacy studies of 2S albumin protein (~12.5 kDa) from *Cucurbita maxima*, coupled with Nano-ZnO at 1:1 molar ratio, showed a potent antimicrobial effect. The 2S albumin-nano-ZnO formulation showed a remarkable decrease in *C*Las population by 96.2% of the initial bacterial load after 30 days of treatment in planta [106].

##### Miraculin-Like Proteins (MLPs)

The native miraculin (24.6 kDa) protein purified from *Richadelladulcifica* (red berries) has a unique taste-modifying property [304]. Miraculin is a homodimer of the glycosylated subunit (191 amino acids long) and cross-linked with a disulfide bridge through Cys138. The proteins, which exhibit 30–50% sequence identity with native miraculin protein, were designated miraculin-like proteins (MLPs) [305]. Both MPLs and miraculin are grouped into the Kunitz-type soybean trypsin inhibitor (STI) family due to sequence identity (30% identity with STI) [306]. Several MLPs characterized from different plant species have been shown to have a role in plant defense and trypsin-inhibitory activity. The two distinct MLPs (I and II) reported in *Citrus jambhiri* showed potential trypsin-inhibitory activity and are involved in plant defense [175]. The phylogenetic studies of MLPs from the soybean Kunitz super family showed that Rutaceae MLP (I and II) clustered on separate branches, and miraculin, along with other MLPs, grouped into distinct clusters. Structure analysis showed that most of the Kunitz-type inhibitors have the same overall fold (β-trefoil fold) and consist of 12 antiparallel β-strands connected by long loops [307]. The 3D structure of MLP from *Murraya koenigii* (MKMLP) has been determined at 2.9 Å resolution (PDB ID: 3IIR). The structural analysis showed that MKMLP is made up of twelve antiparallel β-strands connected through the loop and two short helices. Despite a similar overall fold, a significant difference in the structure was observed in comparison to other reported structures of Kunitz trypsin inhibitors [308]. The serine protease inhibitor of the Kunitz trypsin inhibitor superfamily has been shown to inhibit the serine protease of several lepidopteran insect pests [309].The bioinsecticidal studies of MKMLP against insect pests (*Helicoverpa armigera* and *Spodoptera litura*) have been shown to inhibit the trypsin-like and total protease activity of *Helicoverpa armigera* gut protease (HGP) by 78.5% and 40%, respectively, and *Spodoptera litura* gut protease (SGP) by 81% and 48%, respectively.The prominent proteolytic activity of MKMLP against total insect gut proteinases validates their potential for development as a plant defense agent [310]. In *C*Las-infected sweet orange (*Citrus sinensis*) leaves, noticeable up-regulations of MLP were observed [172].

##### Putranjiva Roxburghii Trypsin Inhibitor (PRTI)

*Putranjiva roxburghii* is an ornamental plant of the *Euphorbiaceae* family. The highly stable and potent trypsin inhibitor from the seed of *Putranjiva roxburghii* was purified and characterized. The PRTI is a single-chain protein with a molecular weight (M_w_) of ~34 kDa that has remarkable stability at a wide range of pH (2–12) and temperature (up to 80 °C). The purified native protein inhibits trypsin with a dissociation constant of 1.4 × 10^−11^ M (at a 1:1 molar ratio). The sequence analysis revealed that it belongs to the Purine Nucleoside Phosphorylase-Uridine Phosphorylase (PNP-UDP) family, and a ~46-residue insert disrupts the PNP domain. The biochemical studies of full length and truncated (without a 46-residue insert) recombinant *Putranjiva roxburghii* PNP family protein (PRpnp) revealed that the active site for trypsin-inhibitory activity is located within the 46-residue insert. The truncated PRpnp (without insert) showed strong PNP enzymatic activity, and the full-length PRpnp showed weak PNP enzyme activity. These results specify the evolution of PRpnp to a potent trypsin inhibitor through a 46-amino-acid-long inhibitory residue to cater to the needs of plant defense [175].

### 5.3. Systemic Acquired Resistance (SAR)

The systemic phenomenon triggered by SA to gain the resistance against pathogen attack, termed as SAR, is also responsible for the expression of defense-related genes [311]. The transgenic-tobacco-expressing SA biosynthesis genes show constitutively the expression of pathogenesis-related (PR) proteins which ultimately lead to resistance against viral and fungal infection [311,312]. Exogenous SA also induces PR genes involved in resistance against viral infection. The induction of SAR, either exogenously or endogenously, would be a potential approach for developing resistance against citrus greening (Figure 10). The defense strategy should be designed to elicit SAR as the pathogens invade the citrus plant. This can be achieved by either applying SA externally or through overexpressing SAR-related genes [191,279]. The NPR1 gene (SAR-inducing proteins) has been used for the induction of the SAR defense system in transgenic citrus. The working principal was an oligomer form of the NPR1 protein that gets converted to a monomeric form after pathogen infection and moves to the nucleus where it activates defense-related genes [313]. The transcription factor induces the expression of the PRgene and ultimately mediates the SA SAR [279]. Transgenic citrus ‘Duncan’ and ‘Hamlin’ lines have been developed by expressing the Arabidopsis NPR1 (AtNPR1) gene against the HLB and citrus canker. The developed lines have displayed an increased tolerance/resistance against HLB and citrus canker [279]. The citrus NPR1 homolog was also overexpressed in ‘Duncan’ grapefruit and reported to activate defense-related genes [283]. A higher level of expression of defense-related genes, i.e., PR1, PR2, and WRKY70, were reported in transgenic sweet orange [314]. The overexpression of the hrpN gene (obtained from *Erwinia amylovora*) in transgenic citrus ‘Hamlin’ also triggered the SAR and hypersensitive response with the consequent reduced severity of citrus canker by 79% [315]. Brassinosteroids activate ZmMPK5, which is involved in self-propagation of apoplastic H_2_O_2_ via regulation of NADPH oxidase gene expression [191]. In brief, there could be a probable correlation between NADPH oxidase gene regulation and the antioxidant defense system of *C*Las. It was also observed that the production of H_2_O_2_ was blocked by pre-treatment with mitogen-activated protein kinase kinase (MAPKK) inhibitors and H_2_O_2_ inhibitors or scavengers. The down-regulation of these inhibitors and scavengers would help for up-regulating the expression of NADPH oxidase. This might be useful in enhancing the immunity of citrus plants against *C*Las by decreasing the amplitude of the antioxidant defense system.

The SA analogues have an ability to block the APX (ascorbate peroxidise) activity to induce defense-related genes in tobacco against tobacco mosaic virus (TMV) [316]. This information would help to down-regulate ascorbate peroxidise via genetic engineering for achieving broad immunity against *C*Las and other pathogens. The degradation of SA by *C*Las salicylate hydroxylase to catechol has been observed [317]. It would be good to develop molecular strategies to down-regulate or inhibit the expression levels of both salicylate hydroxylase and ascorbate peroxidise to enhance/maintain SA levels through design of an antisense cassette for both genes.

## 6. Management of Asian Citrus Psyllid (ACP)

The ACP damages citrus trees directly byfeedingon newly emerged tender leaves (new flush). However, more seriously, the ACP acts as an insect vector of *C*Las and *C*Lam, which are associated with the fatal citrus disease HLB. The psyllid takes the *C*Las bacterium into its body when it sucks the sap of infected citrus plants. The psyllid acts as a carrier of *C*Las and transfers it to the healthy plant. According to reports, bacterial acquisition and multiplication in the nymph and adult are essential for effective transmission [318,319]. Before eventually spreading to plants, *C*Las multiplies and forms a mutually beneficial association with its insect vector [320]. Eggs of the ACP are oval shaped, 0.3 mm long, and yellow/orange in color, which are laid individually or in small clusters, mainly on the tips of growing shoots and also in small cracks on stems. The size of nymphs when they first emerge from the eggs is less than 0.3 mm, which excrete a large quantity of sugary liquid, and they pass through five instars before becoming an adult. External wing pads of the ACP become visible on the later (3rd–5th) instars. The ACP possesses piercing-sucking mouthparts to suck the plant sap as well as inject toxic saliva into the host plant during feeding. Feeding of large numbers of ACPs on newly formed citrus leaves causes the deformation of leaves and shoots and even the death of young flush. Honeydew produced by both nymphs and adults promotes the growth of sooty mold, which covers the leaves and reduces photosynthesis. Injection of salivary toxins during feeding stops terminal elongation and causes the malformation of leaves and shoots.

The new flush on the citrus plant attracts the psyllid females, which lay the eggs, leading to nymph development. In the absence of the new flush on the citrus, psyllid survive on alternative hosts; however, they are metabolically highly active at the temperature ranges from 25 °C to 30 °C. Therefore, the alternative host, i.e., *Murraya paniculata* and *Severinia buxifolia* should be eradicated from citrus groves [321]. Different strategies have been employed for eradication of ACP infestation in the affected citrus groves (Figure 11). The chemical control of the psyllid is the currently preferred strategy, which is highly effective to reduce psyllid populations [322]. The systemic insecticides applied in the soil, like imidacloprid, thiamethoxam, and clothianidin, have been used in large quantities to control psyllids. To achieve the best results, the citrus orchards should be sprayed with effective insecticide just after hedging or topping before initiation of any new flush. The second insecticide spray should be applied after initiation of any new flush to prevent the attack of psyllids. The broad-spectrum insecticides like neonicotinoid, organophosphate, carbamate, and pyrethroid are highly effective against ACP, but they are toxic to other beneficiary insects or natural enemies of psyllids [323].

Natural enemies of any vector (parasitoids, predators, and pathogens) are the main controlling factors which cause the natural mortality in the ecosystem and are some of the basic components of integrated pest management. The ACP can be controlled in the citrus groves using biologically controlling agents, i.e., *Tamarixia radiata* and *Diaphorencyrtus aligarhensis*, which act as parasitoids on *D. citri* (ACP). A number of other entomopathogens like *Isaria fumosorosea*, *H. citriformis*, *Lecanicillium lecanii*, and *Beauveria bassiana* can act as a biopesticide against the ACP [324,325]. The parasitoid *Tamarixia radiata* is the most effective natural enemy that has been successfully used against *D. citri* in various parts of the world [326]. Insecticides like horticultural oil, diflubenzuron, and kaolin clay are most compatible with the *T. radiata* while controlling the psyllid population. Additionally, entomopathogenic fungi *I. fumosorosea* and *B. bassiana* [327] and entomopathogenic bacteria such as *Bacillus thuringiensis* (Bt) [328] have been screened and investigated in recent years. The Bt technology can be considered a potential option in integrated management, as screened Bt strains induced medium to high levels of mortality in *D. citri* 3rd instar nymphs [328]. Alternatively, Bt toxins can be expressed directly in the phloem as bioinsecticides by developing transgenic citrus lines, which could be a promising weapon to control *D. citri* in the war against HLB.

The major predators like spiders, ladybeetles, syrphids, and lacewings that attack the ACP and antifeedants (neonicotinoids, flonicamid, pymetrozine, andimidacloprid), which affect the fertility, are useful to reduce psyllid population in the citrus orchards. The chemical repellents, including noxious plant products, horticultural oil, and physical repellents are complimentary tools for reducing pathogen-transmission-related behaviors [329,330].

The initial recommendation was that symptomatic or diseased trees up to eight years of age had to be destroyed or protected to reduce the pathogen inoculums within an orchard in disease-prone locations. The federal authorities of Brazil recommended (Normative Instruction N°53) that citrus orchards with more than a 28% incidence of symptomatic trees had to be completely eradicated to prevent the spread of HLB [331]. However, most of the citrus producers are cautious of this strategy due to direct fruit loss after eradication of the citrus trees or orchards, thus it is not very well accepted. Considering a scenario in which HLB is present and resistant cultivars are unavailable, there is a need to adopt the strategy that contemplates recommended measures of tree eradication. In order to compensate for the HLB impact and obtain the good citrus yield, citrus growers should concentrate their attention on proper management of citrus orchards. Different ideal conditions have been recommended for citrus grove management, like establishing high-density groves, adequate management of irrigation, and planting varieties with similar phenology. In high-density groves, more citrus plants are planted in order to attain a better yield. The traditional spacing recommendations for a plantation of citrus were 7 to 9 m between rows and 4 to 6 m in the planting row; however, sweet oranges produced well with spacings of 5 to 7 m between rows and 2 to 4 m in the planting row. However, increased planting density in citrus allowed 50% higher harvests in the first eight years than traditional spacings, making it a straightforward tool for enhancing citrus productivity while compensating for the HLB impact [331]. Considering a buffer effect, an orchard with a greater number of trees would be less affected by the tree eradication strategy after infection. It has also been demonstrated that high-density orchards coupled with strict psyllid control had a lower cumulative incidence of HLB over time [332]. If a high-density planting strategy is used around the core orchard, it leads to the control of secondary HLB infections within the citrus groves. This could be due to a “dilution effect” for pathogen inoculum sources arriving via psyllid vector from outside-infected orchards, as high-density orchards have relatively more trees per area than traditional plantings [333]. The orchard can be built by using a higher planting density (up to 200 m) along the farm boundaries to anticipate productivity and reduce HLB impact (Figure 12). The performance of acid lime plantations has been evaluated for two types of density planting systems (high-density, 5 × 2.5 m and ultra-high-density, 2.5 × 2.5 m) and over-conventional (5 × 5 m) planting spacing. High planting density in acid lime helped to maximize the use of cultivated land, light, water, and fertilizer inputs and recorded a two-fold increase in fruit yield (8.24–35.36 t/ha) in the UHD plantation system compared to the control (3.08–11.64 t/ha), implying that more plants can be accommodated into a smaller space to boost production [334]. Currently, the Brazilian citrus planting density has been stabilized around 615–640 trees/ha; the density was increased from 337 to 676 trees/ha. A high-density planting approach with dwarf citrus rootstocks like the ‘Flying Dragon’ trifoliate orange, which has less shoot flush abundance and much lower cumulative HLB incidence than the ‘Rangpur’ lime, can be employed to improve HLB management [333,335].

Planting ACP-repelling citrus hosts, citrus relatives, or other crops around the boundary or as an intercrop may enhance the orchards’ establishment-based HLB management of strategy. Antibiosis (inhibitory) and antixenosis (less attractive) mechanism have been reported in the Poncirus and a few citrus relatives as unfavorable hosts for the ACP [336]. Thus, the breeding approach might be used to develop novel scion cultivars that can repel adult ACP or reduce the survivability of nymphs.

As insecticides have a negative impact on the environment and a severe toxic effect on the other living organisms, including beneficiary insects or natural enemies of the psyllid, there should be an alternative management strategy with the least negative impact on the environment. Nowadays, there are different strategies emerging to control insects. RNA interference (RNAi) is a promising technology that has been used in agriculture to achieve different goals. Furthermore, RNAi is being exploited for the control of insects by silencing the critical gene of the insect through the RNA interference mechanism. This approach can used to inhibit insect growth, increase insecticide susceptibility, affect insect fertility, and cause insect mortality. Tiwari et al. (2011) investigated expression levels of family four cytochrome P450 (CYP4) genes in the *C*Las-infected and uninfected psyllid and reported a higher level of expression of the four genes in uninfected than infected psyllid [337]. They suggested five promising candidate CYP4 genes (CYP4C67, CYP4DA1, CYP4C68, CYP4DB1, and CYP4G70) associated with insecticide resistance in *D. citri* for RNAi-mediated silencing. The silencing of these five genes by topical application of double-stranded RNA leads to the suppression of insecticide resistance in *Diaphorina citri* [338]. The abnormal wing disc (*awd*) gene involved in the development of the wing in the psyllid (instar 5th) has been targeted for silencing. The silencing of the *awd* gene by topical application of dsRNA interfered with wing development, produced malformed wings in adults, and caused significant nymphal mortality [339]. It is reported that a cathepsin-B-like cysteine peptidase (*DCcathB*) gene gets highly expressed (75- to 3333-fold) in the gut of the adult psyllid compared to other tissue. The abundant expression of this gene in the gut may have some correlation with the *C*Las habitat in the gut and therefore is a suggested promising target for HLB control [340]. The muscle protein 20 gene (DcMP20) expresses at a maximum level in the last instar (fourth-fifth) of the nymphal, which isconsidered to be involved in psyllid muscle development. Effort has been made to impair the muscle development in psyllids through RNAi by silencing the DcMP20 gene, which resulted in the significant mortality and reduced body weight in psyllids [341]. Currently, most of the farmers are utilizing heavy doses of insecticide to control the psyllid infestation in citrus orchards, causing the development of insecticide resistance. The development of insecticide resistance in the psyllid can be minimized by silencing the genes that are responsible for gaining resistance. Kishk et al. (2017) used the RNAi approach to silence the genes carboxyesterases (*EstFE4*) and acetylcholinesterases (*AChe*) implicated in pesticide resistance in order to increase the susceptibility of the psyllid towards the insecticides by topical application of the dsRNA [342]. They observed that a topical application of the dsRNA caused concentration-dependent nymph mortality.

Advanced research on RNAi indicates that the knockdown of target genes chitin synthase, cathepsin D, andgenes inhibiting apoptosis can be carried out through oral feeding of dsRNA as well as through topical feeding. Ultimately, it causes mortality in nymphs and adults of *D. citri* [343,344]. Developing transgenic citrus plants expressing antisense RNA or direct application of dsRNA of critical genes involved in psyllid growth/metabolism/skeleton could serve as a potential approach for psyllid and HLB management. Besides that, the silencing of targeted genes can also be carried out through the infectious viral-based vector system. The major advantage of the viral-based vector system is that it is faster than the transgenic approach for RNAi. Citrus tristeza virus (CTV) can be used as a transient expression vector to control the HLB disease because of its stability. A viral vector could act as a gun and RNAi and antimicrobial peptides could act as bullets to fight against disease [345,346,347,348]. The CTV-based vector can be used simultaneously against *C*Las by expressing the antimicrobial peptides and psyllid through RNAi-mediated silencing to mitigate the HLB in citrus plants.

The sterile insect technique (SIT)/self-limiting genetic technology is one of the promising control aspectsthat could be used to decrease the insect population, in which the reproductive process of the vector insect is disrupted [349,350]. Such sterile insects can be used to control their own population by releasing them in the environment. The eggs produced by native females mating with sterile insects will not hatch. Most of the females will produce sterile eggs, ultimately causing the ratio of sterile to normal insects to increase, and the native insect population will become extinct [351,352]. This approach includes the release of only sterile male insects near the commercial citrus production areas, but there is no successful report of SIT application for ACP. However, at the University of California, the research is ongoing to take the initiative on SIT application to control the psyllid population [353].

## 7. Conclusions

Several efforts have been undertaken to effectively manage HLB around the world. In the absence of a cure, most citrus producers have chosen to live with infected trees rather than eradicating them. Antimicrobial compounds (such as antibiotics and nanoparticles), potential resistance inducers, phytohormones, thermotherapy, improved nutritional programming (ENP), and symptomatic branch trimming are among the disease control strategies that have been evaluated to mitigate disesase impact. Although a couple of these approaches have shown to be beneficial for combating the disease, it would be prohibitively expensive to implement on a broad scale in the field. Therefore, there is a need to further evaluate disease control strategies by targeting the pathogen spread at three different levels. (i) Pathogen (a) Targeting and inhibiting an important proteins of *C*Las, like transport proteins (ABC transporter, Znu system, amino acid uptake system, Sec-translocase/translocon), transcription regulators, hydrolase family enzymes (phosphatase, serine protease), proteins involved in antioxidant defense system, proteins involved in nucleotide biosynthesis (ATIC, Inosine-5′-monophosphate dehydrogenase), proteins involved in fatty acid biosynthesis (FabI, FabZ), and proteins involved in amino acid biosynthesis. (b) Use of the most efficient antimicrobial or immunity-inducing compound to suppress the growth of *C*Las, viz., antibiotics, brassinosteroids, salicylic acid, stable antimicrobial peptides (SAMPs), and engineered nanomaterials. (c) Use of methodology to kill the *C*Las, viz., thermotherapy and cryotherapy. (ii) Host (a) Improvement of host immune system, (b) Effective use of transgenic approach to build the host resistance against *C*Las, (c) Induction of systemic acquired resistance. (iii) Vector (a) Chemical control, (b) Biological control, (c) Adopt the strategy that contemplates recommended measures of trees’ eradication (proper citrus orchard management, use of a high-density planting approach, planting ACP-repelling trees), (d) Use of advanced technology (RNAi, genome-editing technology, sterile insect technique technology).

Here, we introduced triangular disease management strategies by keeping the checkpoints for pathogen spread at three different levels. In order to establish HLB-free citrus orchards, several check points or obstacles should be implemented to prevent pathogen spread to a healthy orchard. The first line of defense includes the selection of naturally tolerant mother plants, the use of disease-free planting material for propagation during the establishment of the orchard at the initial stage, and cultural management approaches to avoid infection from other diseases (Figure 13A,B). The quest for a disease-resistant citrus cultivar and the integrated triple action (healthy nursery trees, orchard surveillance and infected tree destruction, and vector control) are important subjects for mitigating the adverse impact of HLB. Thus, during the establishment of new citrus orchards, there is a need to use psyllid-repellent or suitable intercropping plants or border plants. There are various alternative hosts that have been reported for psyllid. Therefore, alternative hosts should be removed from the orchard. An HLB-free citrus belt should be saved from the introduction of a new pathogen through effectual quarantine measures for the host or alternative host. Alternatively, vectors can be managed by applying different chemicals in the orchard after the appropriate duration to avoid *D. citri* infestation. The orchards should be under inspection and examined regularly for the psyllid/disease, and if any symptomatic tree is observed in the orchard, the affected trees should be isolated using insect-proof nets or destroyed. The pruning of effected trees is also beneficial to avoid disease spread. Management of vector population only by insecticide application is a challenging job. To address such challenges, several research strategies are being explored, including the investigation of the mechanism of the acquisition of *C*Las and antagonistic microorganisms by *D. citri*, with the goal of eliminating or limiting pathogen colonization in the vector [354].

The proper application of enhanced nutritional programs (ENPs) in the citrus orchard to reduce HLB symptoms is important. The ENPs’ formulation containing essential micronutrients, such as phosphate and salicylate, is shown to reduce nutrient deficiency symptoms [355]. A high-density planting with dwarf citrus rootstocks like the ‘Flying Dragon’ trifoliate orange, which has much lower cumulative HLB incidence than the ‘Rangpur’ lime, can be employed to compensate for the HLB impact [333,335]. It was speculated that the intercropping of guava plants in citrus orchards reduces infestations of the psyllid population, termed the “guava effect” on HLB [356]. Certain volatile compounds which are present in guava plants are shown to inhibit/block the psyllid response towards the citrus odor. The volatile compound reported in the guava is (E)-β-caryophyllene, which has an ACP-repellent property, leading to low HLB incidence in groves. Mango (*Mangifera indica*) is another plant that serves as a psyllid deterrent, whereas orange jasmine (*Murraya paniculata*) is the most attractive host for the psyllid. The appropriate utilization of such promising intercrops results in just 3–20% HLB incidence, compared to 76% incidence in citrus without intercropping [333]. As a result, proper planting of both psyllid-repellant and attractive plants across the entire grove may reduce the HLB effect.

To ensure sustainable citriculture, obtaining a resource of genetic resistance for *C*Las is a prerequisite. However, no resistant citrus genotype has been reported, with the exception of a few Rutaceae family citrus relatives, the subfamily Aurantioideae, which are incompatible with citrus. The researchers are aggressively looking for *C*Las-resistant graft-compatible genotypes.It was reported that *Citrinae* species did not allow bacterial proliferation (*Microcitrus* and *Eremocitrus*). These sources could be used in breeding programs to develop new *C*Las-resistant citrus rootstocks [203]. The development of *C*Las-resistant citrus rootstocks by hybridization with graft-compatible genotypes would be a significant step in combating HLB.

As a result of the unavailability of natural resistance in citrus and similar genera, the genetic engineering approach for the development of an HLB-resistant commercial variety would have a high chance of success [2]. Transgenic citrus plants having an inherent ability to emit (E)-β-caryophyllene could potentially be used as new protection strategies for citrus trees from psyllid as well as HLB [357]. The release of exogenous dsRNAs on the citrus plant to silence/shutdown the essential genes of insects using the RNA interference technique has considerable potential in insect pest management, particularly psyllid control [347]. Allowing psyllids to feed on antisense dsRNA-coated citrus plant leaves or transgenic citrus plants producing dsRNA can trigger the RNAi process.

A hypothetical model depicted below step-by-step could potentially save the citrus industry from the HLB pandemic (Figure 14). (i) Choose the mother plants having desired horticultural traits and index them for all virus and virus-like pathogens (including *C*Las) using molecular techniques (ii) Use healthy mother plantsfor the production of large-scale planting material using the meristem tissue culture technique. (iii) Treat the apical meristem tissue with thermotherapy, cryotherapy, and chemotherapy for obtaining pathogen-free explants. (iv) Use such explants for the production ofa large number of pathogen-free planting materialsby plant tissue culture techniques. (v) Supply hardened planting material with appropriate nutrition. (vi) Introduce these pathogen-free trees to the newly designed orchard with proper intercropping (such as guava, garlic, soybean) to avoid psyllid invasion or preferably protect with a psyllid-attractive border crop. (vii) Manage the orchard with multipronged chemical and biological tools, which are backed by strong scientific evidence.

## Figures and Tables

**Figure 1 plants-12-00160-f001:**
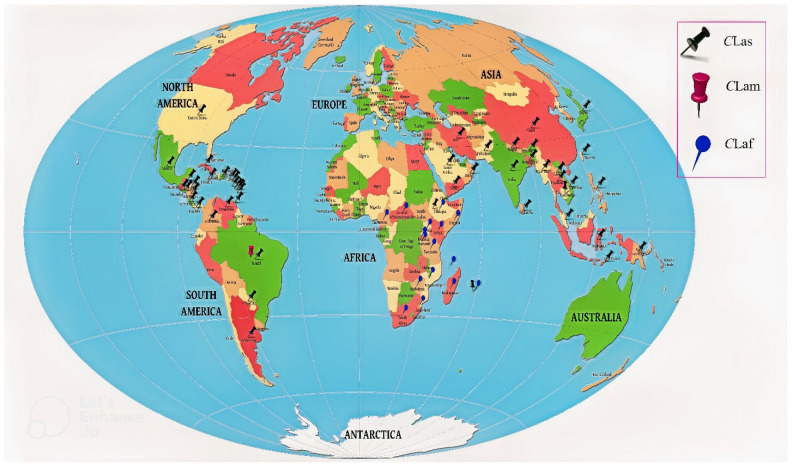
The world map represents geographical distribution of HLB based on DNA-DNA hybridization with probe, PCR followed by *Xbal* restriction digestion of the amplified DNA, electron microscopy, and symptomatology.

**Figure 2 plants-12-00160-f002:**
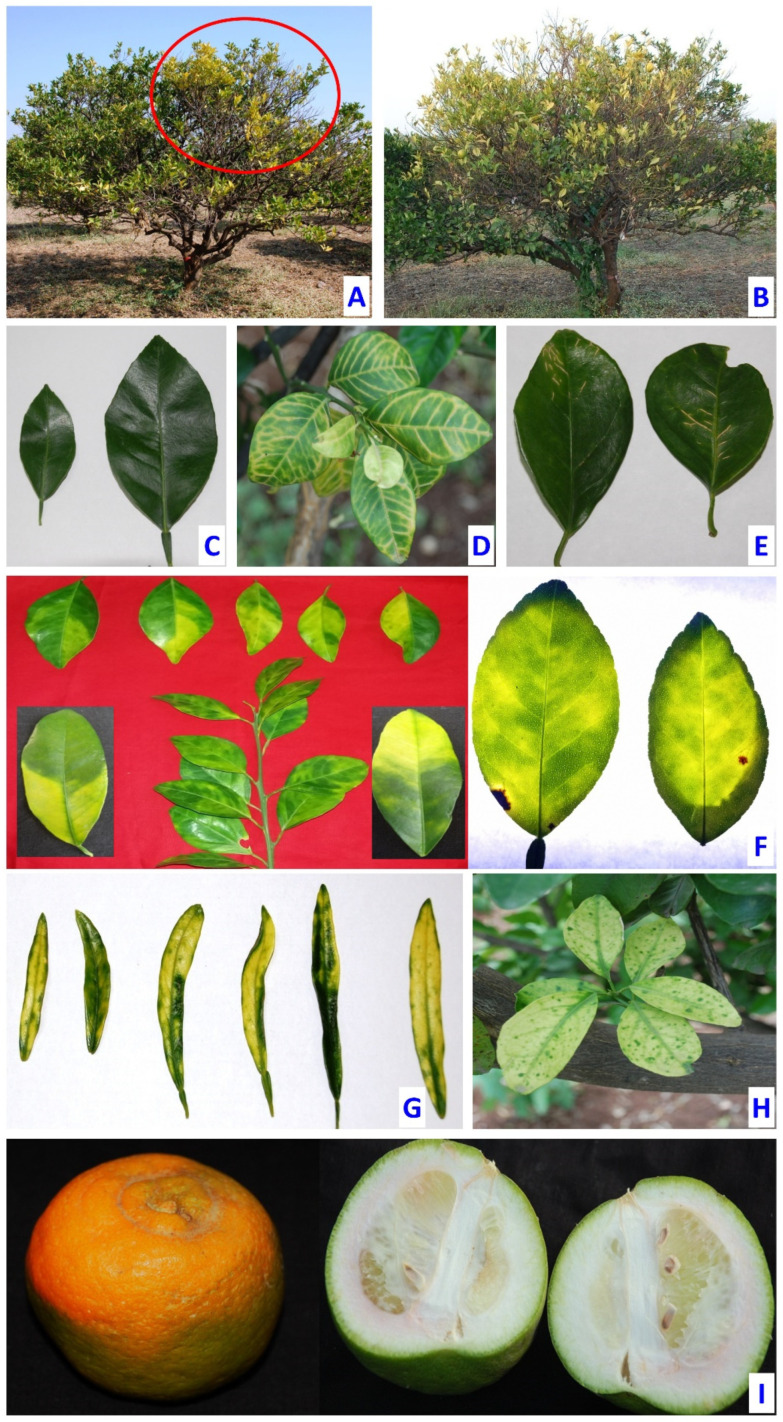
(**A**) HLB-infected sweet orange plant in the field showing characteristic yellow shoot symptoms at initial stage. (**B**) HLB-infected sweet orange in the field at severe stage. (**C**) Healthy leaf. (**D**) Vein yellowing and corking. (**E**) Vein corking. (**F**) Blotchy mottle (a random pattern of chlorosis). (**G**) Narrow leaf with blotchy mottle. (**H**) Green island. (**I**) HLB-affected with color inversion and misshapen sweet orange fruit (lopsided) with aborted seeds.

**Figure 3 plants-12-00160-f003:**
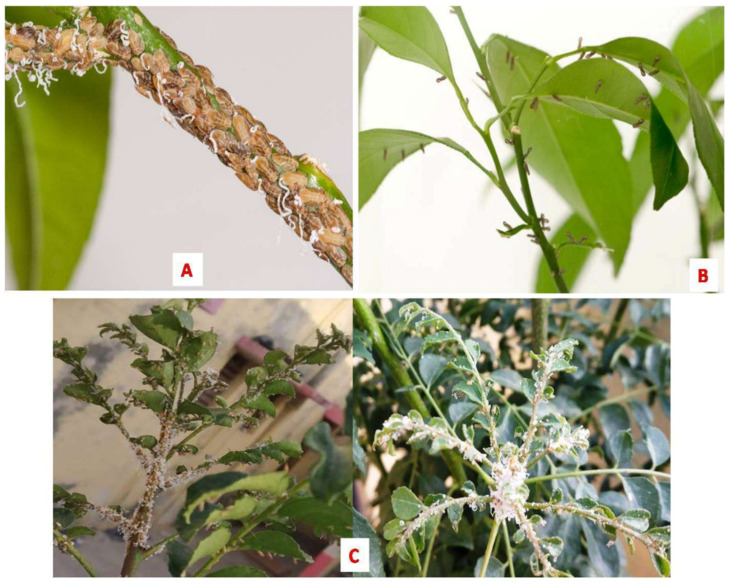
(**A**) ACP nymph on a citrus plant in the field. (**B**) ACP adults feeding on the leaf. (**C**) Infestation of ACP nymph and adults on curry leaf plants.

**Figure 5 plants-12-00160-f005:**
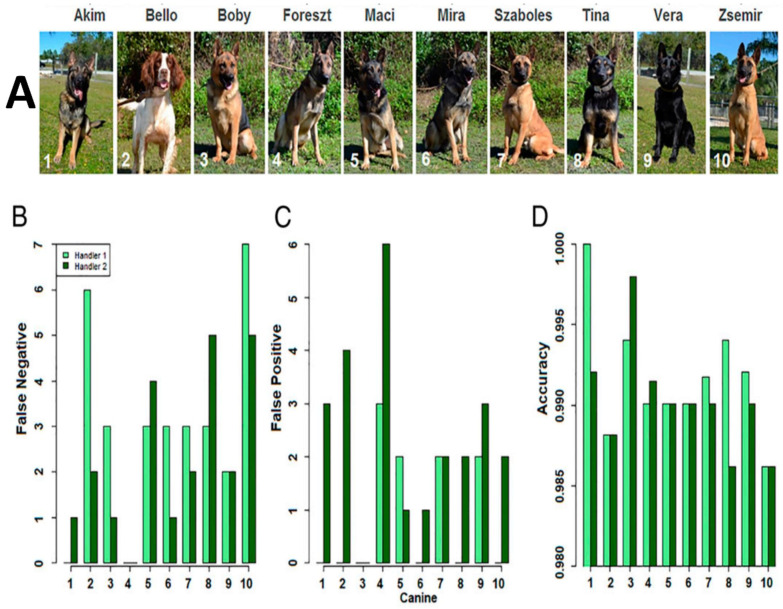
Canine performance for detecting *C*Las in citrus. (**A**) The canines with the fewest total errors were Akim, Boby, and Mira with four, six, and six errors, respectively. Caninehandler team performance assessment. (**B**) False negative error. (**C**) False positive error. (**D**) Overall accuracy [118].

**Figure 6 plants-12-00160-f006:**
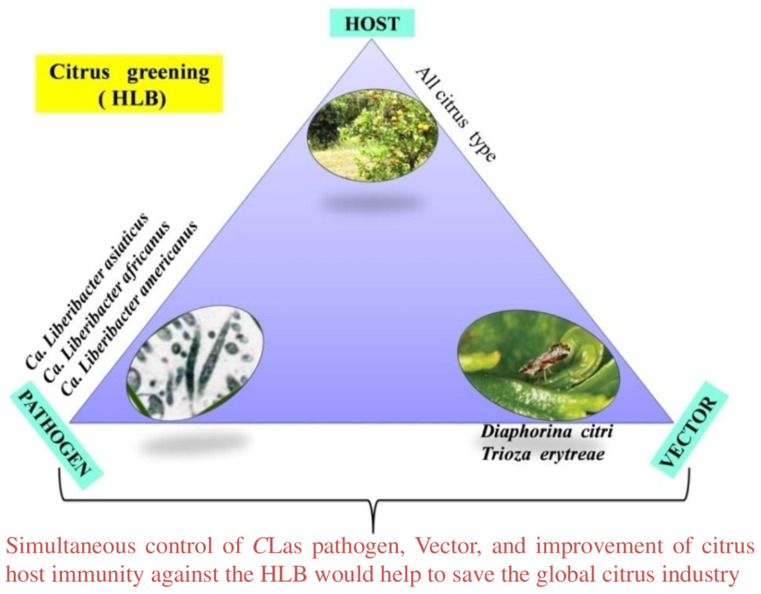
Triangular control management of HLB.

**Figure 7 plants-12-00160-f007:**
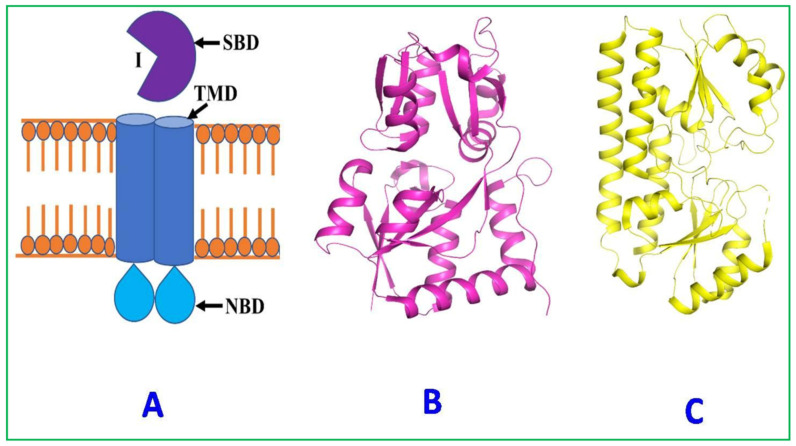
(**A**) Schematic representation of ABC transporter complexes with SBDs in Gram-negative bacteria. (**B**) Crystal structure of periplasmic cystine-binding protein from *C*Las (PDB Id: 6A80). (**C**) Crystal structure of periplasmic metal-binding protein from *C*Las (PDB Id: 4Cl2). SBD: Solute-binding domain, TMD: Transmembrane domains, NBD: Nucleotide binding domain.

**Figure 8 plants-12-00160-f008:**
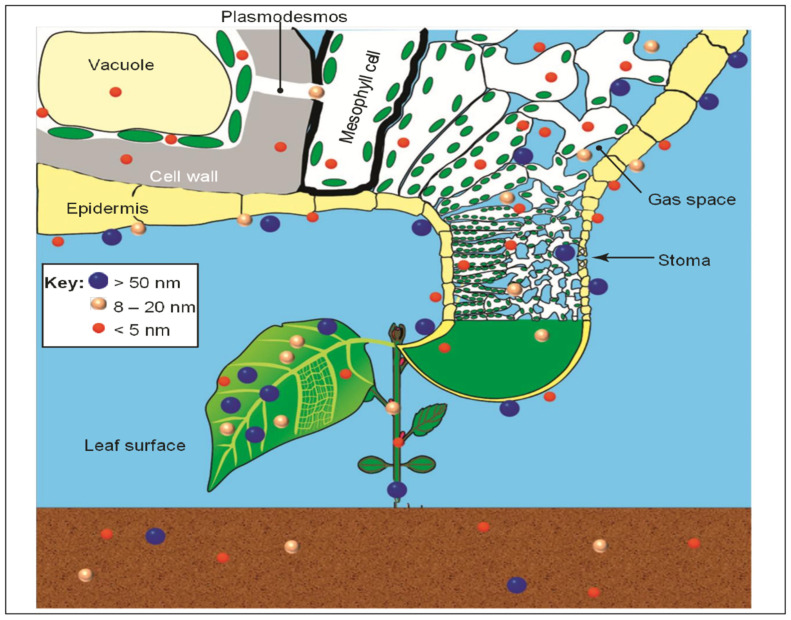
Entry of different sizes of nanoparticles to plant vascular system and mobilization between tissues (adapted from Dietz et al., 2011; copyright 2011 Elsevier) [216].

**Figure 9 plants-12-00160-f009:**
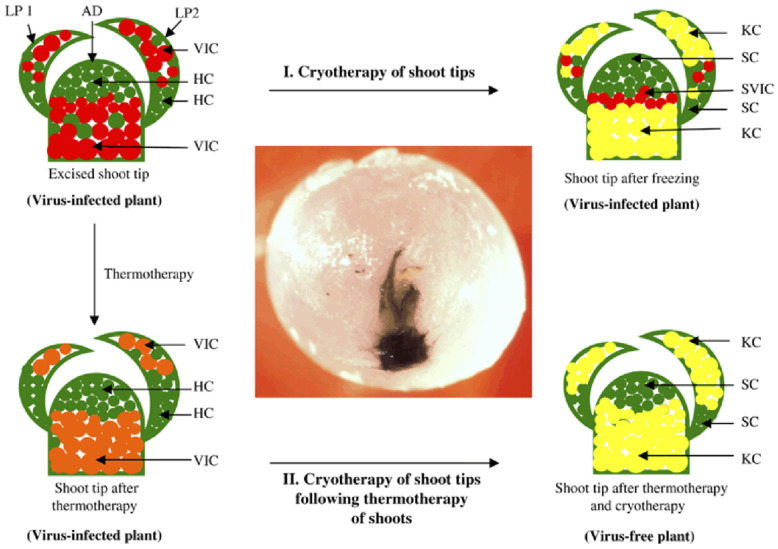
The schematic representation of combination of thermotherapy and cryotherapy for enhanced elimination of pathogen (*C*Las) that are capable of invading the meristematic cells. AD, apical dome; HC, healthy cells; KC, killed cells; LP1, leaf primordium1 (youngest); LP2, leaf primordium2; SC, surviving cells; SVIC, surviving, virus-infected cells; VIC, virus-infected cells [267].

**Figure 10 plants-12-00160-f010:**
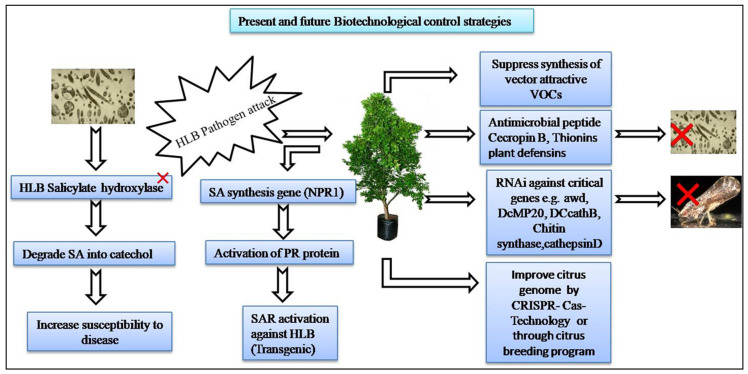
Schematic representation of biotechnological strategies to control greening disease.

**Figure 11 plants-12-00160-f011:**
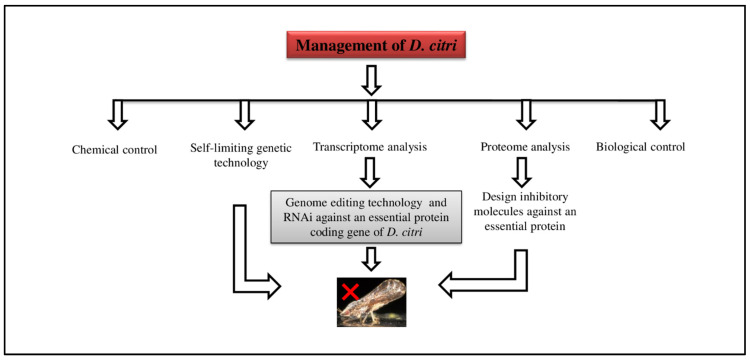
Different strategies to manage the ACP population.

**Figure 12 plants-12-00160-f012:**
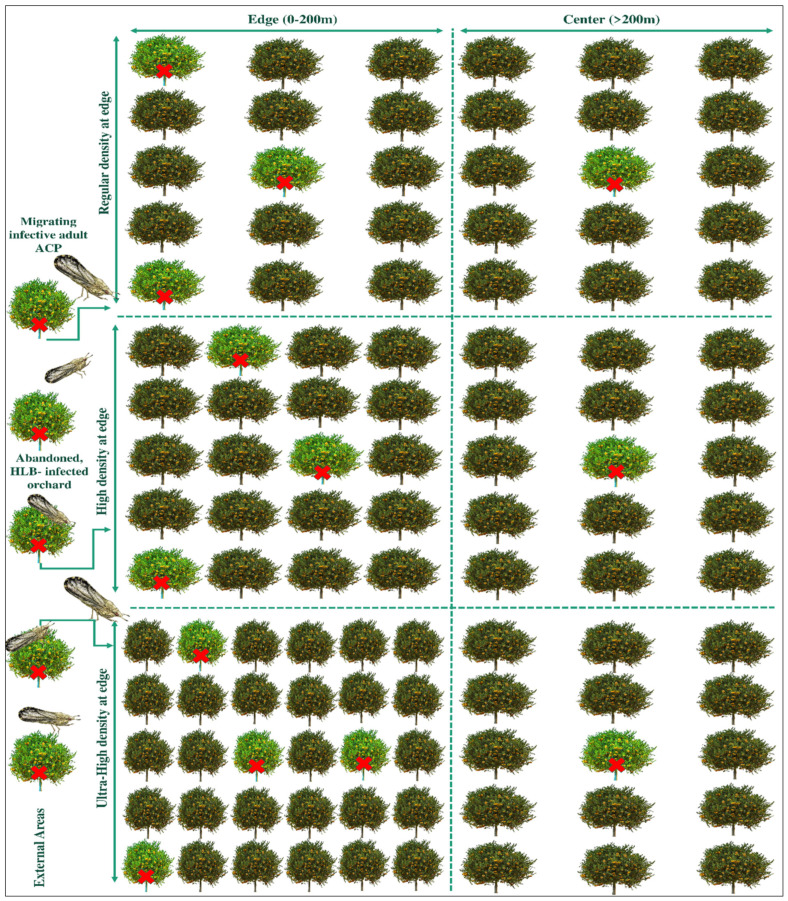
The effect of high tree density along the edge of well-managed orchards on citrus greening spread is depicted schematically. The orchards are mostly affected from the outside by a carrier psyllid vector. The high planting around the orchards may act as a temporary barrier for movement of ACP, resulting in a larger ratio of healthy to infected trees (dilution effect).

**Figure 13 plants-12-00160-f013:**
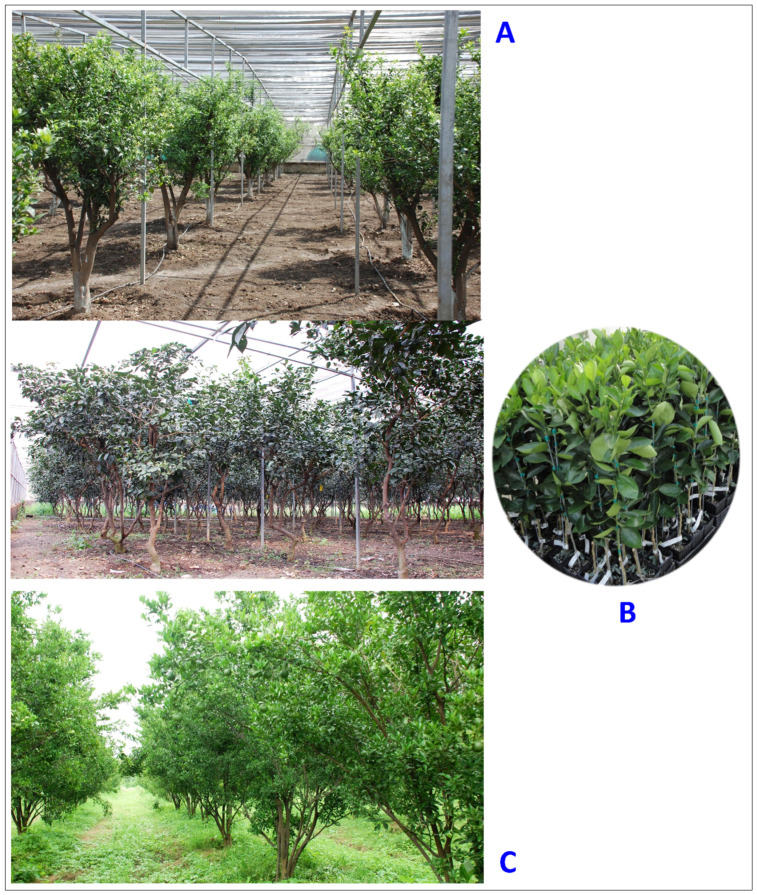
(**A**). Protected mother plants of Nagpur mandarin (Source of HLB and Virus-free scion) (**B**). Certified HLB-free-quality nursery citrus plants (**C**). Well-established healthy orchard of Nagpur mandarin.

**Figure 14 plants-12-00160-f014:**
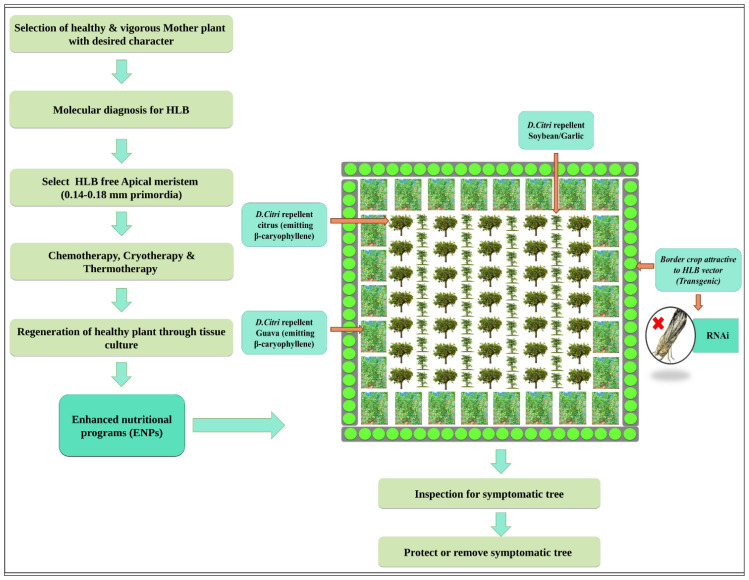
Overall preventive measures to establish HLB-free citrus orchards.

**Table 1 plants-12-00160-t001:** Worldwide distribution of HLB disease.

Sr. No	Continent/Country/Region	Distribution of HLB	Causal Organism	Reference
	Asia			
1	Bangladesh	Present	*Ca*. Liberibacter asiaticus	[21]
2	Bhutan	Present	*Ca*. Liberibacter asiaticus	[22]
3	Cambodia	Present	*Ca*. Liberibacter asiaticus	[23]
4	China	Present	*Ca*. Liberibacter asiaticus	[22]
5	India	Present, Widespread	*Ca*. Liberibacter asiaticus	[24]
6	Indonesia	Present	*Ca*. Liberibacter asiaticus	[25]
7	Iran	Present, Localized	*Ca*. Liberibacter asiaticus	[26]
8	Japan	Present	*Ca*. Liberibacter asiaticus	[27]
9	Laos	Present	*Ca*. Liberibacter asiaticus	[23]
10	Malaysia	Present, Localized	*Ca*. Liberibacter asiaticus	[24]
11	Myanmar	Present	*Ca*. Liberibacter asiaticus	[23]
12	Nepal	Present, Widespread	*Ca*. Liberibacter asiaticus	[28]
13	Oman	Present, Localized	*Ca*. Liberibacter asiaticus	[22]
14	Pakistan	Present	*Ca*. Liberibacter asiaticus	[29]
15	Philippines	Present, Widespread	*Ca*. Liberibacter asiaticus	[29]
16	Saudi Arabia	Present, Localized	*Ca*. Liberibacter asiaticus	[30]
17	Sri Lanka	Present	*Ca*. Liberibacter asiaticus	[22]
18	Taiwan	Present, Widespread	*Ca*. Liberibacter asiaticus	[22]
19	Thailand	Present	*Ca*. Liberibacter asiaticus	[31]
20	Vietnam	Present, Localized	*Ca*. Liberibacter asiaticus	[32]
21	Yemen	Present, Localized	*Ca*. Liberibacter asiaticus	[30]
	North America		*Ca*. Liberibacter asiaticus	
22	Barbados	Present, Localized	*Ca*. Liberibacter asiaticus	[22]
23	Belize	Present, Localized	*Ca*. Liberibacter asiaticus	[17]
24	Costa Rica	Present, Localized	*Ca*. Liberibacter asiaticus	[22]
25	Cuba	Present, Widespread	*Ca*. Liberibacter asiaticus	[22]
26	Dominica	Present, Few occurrences	*Ca*. Liberibacter asiaticus	[22]
27	Dominican Republic	Present, Localized	*Ca*. Liberibacter asiaticus	[22]
28	El Salvador	Present	Unknown	[33]
29	Guadeloupe	Present, Localized	Unknown	[34]
30	Guatemala	Present	*Ca*. Liberibacter asiaticus	[22]
31	Honduras	Present	*Ca*. Liberibacter asiaticus	[22]
32	Jamaica	Present, Widespread	*Ca*. Liberibacter asiaticus	[22]
33	Martinique	Present, Localized	*Ca*. Liberibacter asiaticus	[34]
34	Mexico	Present, Localized	*Ca*. Liberibacter asiaticus	[22]
35	Nicaragua	Present	*Ca*. Liberibacter asiaticus	[22]
36	Panama	Present, Localized	*Ca*. Liberibacter asiaticus	[22]
37	Puerto Rico	Present	*Ca*. Liberibacter asiaticus	[22]
38	Trinidad and Tobago	Present, Localized	*Ca*. Liberibacter asiaticus	[22]
39	U.S. Virgin Islands	Present, Few occurrences	*Ca*. Liberibacter asiaticus	[22]
40	United States	Present, Localized	*Ca*. Liberibacter asiaticus	[22]
	South America			
41	Argentina	Present, Localized	*Ca*. Liberibacter asiaticus	[22]
42	Brazil	Present, Localized	*Ca*. Liberibacter americanus and Ca. Liberibacter asiaticus	[9]
43	Colombia	Present, Few occurrences	*Ca.*Liberibacter asiaticus	[22]
44	Paraguay	Present, Localized	*Ca*. Liberibacter asiaticus	[35]
45	Venezuela	Present	*Ca*. Liberibacter asiaticus	[36]
	Africa			
46	Burundi	Present	*Ca*. Liberibacter africanus	[37]
47	Cameroon	Present	*Ca*. Liberibacter africanus	[37]
48	Central African Republic	Present	*Ca*. Liberibacter africanus	[37]
49	Comoros	Present	*Ca*. Liberibacter africanus	[22]
50	Eswatini	Present	*Ca*. Liberibacter africanus	[38]
51	Ethiopia	Present	*Ca*. Liberibacter africanus and *Ca*. Liberibacter asiaticus	[37]
52	Kenya	Present	*Ca*. Liberibacter africanus	[29]
53	Madagascar	Present	*Ca*. Liberibacter africanus	[38]
54	Malawi	Present	*Ca*. Liberibacter africanus	[37]
55	Mauritius	Present	*Ca*. Liberibacter africanus and *Ca*. Liberibacter asiaticus	[22]
56	Rwanda	Present	*Ca*. Liberibacter africanus	[37]
57	Somalia	Present	*Ca*. Liberibacter africanus	[22]
58	South Africa	Present, Localized	*Ca*. Liberibacter africanus	[39]
59	Tanzania	Present, Localized	*Ca*. Liberibacter africanus	[38]
60	Uganda	Present	*Ca*. Liberibacter africanus	[40]
61	Zimbabwe	Present, Localized	*Ca*. Liberibacter africanus	[29]

**Table 3 plants-12-00160-t003:** Binding studies of a solute-binding protein from different bacteria.

Protein		Organism	PDB Id	Ligand	Dissociation Constant (K_D_)	References
ArtJ		*Geobacillus stearothermophilus*	2Q2A	Arginine	0.039 µM	[137]
			2Q2C	Histidine	0.42 µM	
			2PVU	Lysine	0.49 µM	
GlnH		*Mycobacterium tuberculosis*	6H2T	Glutamate	15.2 µM	[138]
			6H1U	Aspartate	4.8µM	
			6H20	Aspargine		
DEBP		*Shigella flexneri*	2VHA	Glutamate	1.6 µM	[139]
				Aspartate	5.2 µM	
GlnP		*Lactococcuslactis*	4KQP	Glutamine	1.49 µM	[140]
CjaA		*Campylobacter jejuni*	1XT8	Cysteine	100 µM	[141]
GlnBP		*Escherichia coli*	1WDN	Glutamine	0.5µM	[127]
HBP		*Escherichia coli*	1HSL	Histidine	0.064 µM	[142]
HisJ		*Salmonella typhimurium*	1HPB	Histidine	0.030 µM	[143]
LAO		*Salmonella typhimurium*	1LST	Lysine	0.014 µM	[143]
Ngo0372		*Neisseria gonorrhoeae*	2YLN	Cystine	0.021 µM	[144]
Ngo2014		*Neisseria gonorrhoeae*	2YJP	Cysteine	0.026 µM	[144]
CLasTcyA		*Candidatus* Liberibacter asiaticus	6A80	Cystine	1.26 µM	[145]
ClasTcyA Mutant (V58W)		*Candidatus* Liberibacter asiaticus	-	Cystine	0.22 µM	[146]
CLas-ZnuA2		*Candidatus* Liberibacterasiaticus	4UDO	Mn^2+^	370 µM	[147,148]
			5AFS	Zn^2+^	430 µM	
			6IXI	Cd^2+^		[146]
CLas-ZnuA2 Mutant	(S38A)	*Candidatus* Liberibacterasiaticus	5Z2K	Mn^2+^	340 µM	[149]
	(Y68F)	*Candidatus* Liberibacterasiaticus	5ZHA	Mn^2+^	540 µM	
LBP		*Escherichia coli*	1USK	Leucine	0.4 µM	[150]
			1USI	Phenylalanine	0.18 µM	

**Table 4 plants-12-00160-t004:** ABC transporter system of *C*Las.

GenBank Accession No ^£.^	Super Family	Species		Domain	Putative Function	*C*Las Relative Expression ^$^
		Query Coverage ^€^	Identity ^€^			
ACT56612	MlaF	*Acinetobacter baumanii*		NBD	Putative ATP-binding component of ABC transporter: Involved in resistance to organic component	7.60
		92%	38.49%			
ACT56643	PBP2_ BztA	*Brucellaovis*		SBD	Putative cationic amino acid ABC transporter	−1.29
		93%	57.32%			
ACT56645 (*aapM*)	TM_ PBP2	*Caldanaerobactersubterraneus*		Permease	Putative general L-amino acid transport system permease	3.64
		54%	30.77%			
ACT56815 (*proX*)	PBP2_ ChoX	*Sinorhizobiummeliloti*		SBD	Putative glycine betaine/proline ABC transporter	1.84
		89%	50.0%			
ACT56816	FieF	*Escherichia coli*		Efflux protein	Predicted cation (Co/Zn/Cd) efflux transporters	3.88
		91%	26.83%			
ACT57010 (ZnuA2)	TroA-like	*Yersinia pestis*		SBD	Fi/Mn transport	7.92
		92%	49.26%			
ACT57013	ZnuB	-		Permease	Putative Mn^2+^/Zn^2+^ transport system	5.72
		-	-			
ACT57585	PBP2	*Neisseria gonorrhoeae*		SBD	General L-amino acid Transport	2.97
		83%	32.33%			
ACT57586	TM_ PBP2	*Caldanaerobactersubterraneus*		Permease	ABC-type amino acid transport system	7.96
		83%	34.04%			
ACT57180	PBP2	*Planctopiruslimnophila*		SBD	Putative periplasmic phosphate-binding protein	1.28
		82%	29.01%			
ACT57178	PstA	-		Permease	Putative phosphate transporter	2.23
		-	-			
ACT57369	livM	-		Permease	Putative branch chain aminoacid (Leucine, Isoleucine, Valine) transporter	-
		-	-			

^£.^ Accession numbers are for ABC transporter in the *C*Las Psy62 genome. ^€^ Query coverage and percent identity shows protein BLAST against protein data bank. ^$^ Fold change (log2 ratio) is the relative gene expression (in plantaversus in psyllid) of *C*Las. Positive value indicates overexpressed in planta and negative value showed overexpression in psyllid.

**Table 5 plants-12-00160-t005:** Transcription regulators of *C*Las.

GenBank Accession No	Regulator Type	Protein Name	Putative Function	Mw (kDa)	*C*Las Relative Expression ^$^	References
ACT56890	CarD	PrbP	Regulate some ribosomal gene expression (Activator)	21.3 Kda	1.24	[159]
ACT56824	MarR	LdtR	Activator and Repressor of gene expression	19.6 Kda	5.23	[160]
ACT57167	LuxR	VisN	Activator of chemotaxis, flagellar, and motility genes.	28.5 Kda	2.75	[161]
ACT56755	LysR	LsrB	Activator of oxidativestress lipopolysaccharide biosynthesis gene.	34.3 Kda	2.23	[163]
ACT56897	HTH-XRE	PhrR1	Gene related to quorum sensing	16.5 Kda	Not reported	[162]
ACT57366	Response regulator	CtrA	Control cell cycle	26.7 Kda	1.4	[164]
ACT57093	IscR	RirA	Response to iron limitation	16.1 Kda	1.04	[165]

^$^ Fold change (log2 ratio) is the relative gene expression (in plantaversus in psyllid) of *C*Las. Positive value indicates overexpressed in planta and negative value showed overexpression in psyllid.

**Table 6 plants-12-00160-t006:** Crystal structure of important protein from *C*Las.

Protein	GenBank Accession No	PDB ID	References
Periplasmic metal-binding protein	ACT57010	4CL2, 4UDN, 4UDO, 5AFS, 5Z2J, 5Z2K, 5Z35, 5ZHA, 6IXI	[146,147,148,149]
Putative amino-acid-binding periplasmic ABC transporter protein	ACT57585	6A80, 6AA1, 6AAL, 6A8S	[145]
Enoyl-Acyl Carrier Protein Reductase I (FabI)	KAE9510327	4NK4, 4NK5	[183]
beta-Hydroxyacyl-acyl carrier protein dehydratase (FabZ)	WP_015452391	4ZW0	[185]
Inosine 5′-monophosphate Dehydrogenase	ACT57362	6KCF	[187]

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
