# Peer review of "Huanglongbing Pandemic: Current Challenges and Emerging Management Strategies"

_plants, 2022, doi:10.3390/plants12010160_

Round 1
Reviewer 1 Report
This ms. reviews and elaborates extensively various methods and strategies used or suggested for combatting HLB pathogens in the citrus host, or in the psyllid vector. The authors did a good job in describing and illustrating various possible control strategies. My main criticism of the ms. is that the authors while describing certain areas in too much details (e.g. mehods of transmission electron micoscopy and molecular techniques (page13), they oversimplified the complicated relationship between the pathgen and the vector. Clas does not act merely as a symbiont of the vector D. citri (line 106), and the latter is not merely a carrier of the pathogen (line 1338). Several investigations (3 examples below) show that Clas reproduces in this vector and affects its biology both positively and negatively. Authors should not ignore these and other important publications that show a more complicated pathgen-vactor relationship:
Inoue et al. 2009. Ann. Appl. Biol. 155.
Ammar et al. 2016. PlosOne. 11.
Pelz-Stelinsky and Killiny 2016. Ann. Entom. Soc. Amer. 109.
Reviewer 2 Report
The review"Huanglongbing pandemic: current challenges and emergingmanagement strategies" by Dilip Ghosh et al. is an interesting work
that presents up-to-date literature in the field.
I have no comments and recommend publication in its current form.
-line 395 only change Traingular with Triangular
